# LETO: MODELING MULTIVARIATE TIME SERIES WITH MEMORIZING AT TEST TIME

## ABSTRACT

Modeling multivariate time series data has been at the forefront of machine learning research efforts across diverse domains. However, effectively capturing dependencies across both time and variate dimensions, as well as temporal dynamics, have made this problem extremely challenging under realistic settings. The recent success of sequence models, such as Transformers, Convolutions, and Recurrent Neural Networks, in language modeling and computer vision tasks, has motivated various studies to adopt them for time series data. These models, however, are either: (1) natively designed for a univariate setup thus missing the the rich information that comes from the inter-dependencies of time and variate dimensions; (2) inefficient for long-range time series; and/or (3) propagating the prediction error over time. In this work, we present LETO, a native 2-dimensional memory module that takes the advantage of temporal inductive bias across time while maintaining the permutation equivariance of variates. LETO uses meta in-context memory modules to learn and memorize patterns across the time dimension, and simultaneously, incorporates information from other correlated variates, if needed. Our experimental evaluation shows the effectiveness of LETO on extensive and diverse benchmarks, including time series forecasting (short, long, and ultra-long), classification, and anomaly detection.

## 1 INTRODUCTION

Modeling multivariate time series data is a well-established problem in the literature with a diverse set of applications ranging from healthcare (Ivanov et al., 1999; Tang et al., 2023) and neuroscience (Behrouz & Hashemi, 2024a) to finance (Gajamannage et al., 2023; Pincus & Kalman, 2004), energy (Zhou et al., 2021), transportation management (Durango-Cohen, 2007), and weather forecasting (Allen et al., 2025; Price et al., 2025). Classical shallow models—such as State Space Models (Harvey, 1990; Aoki, 2013), ARIMA (Bartholomew, 1971), SARIMA (Bender & Simonovic, 1994), Exponential Smoothing (ETS) (Winters, 1960)—have long been the de-facto mathematical models for time series prediction, modeling diverse complex patterns (such as seasonal and trend patterns). Deploying these models at scale in real-world settings remains challenging due to their reliance on manual data preprocessing, sensitive model selection, and inherently sequential, non-parallelizable computations. Additionally, these models often fail to capture (1) the inter-dependencies of different variates, and (2) the complex *non-linear* dynamics inherent to multivariate time series data.

The emergence of deep learning has shifted the focus of recent time series research away from traditional statistical methods toward deep neural network architectures such as Transformer-based (Zhou et al., 2021; Wu et al., 2021), recurrence-based (Behrouz et al., 2024d;e; Patro & Agneeswaran, 2024; Jia et al., 2023), and temporal convolutional-based (Bai et al., 2018; Sen et al., 2019; Luo & Wang, 2024) models. Despite the outstanding performance of Transformers (Vaswani et al., 2017) across various diverse domains (Du et al., 2023; Nguyen et al., 2024; Wu et al., 2021), recent studies have highlighted their frequent suboptimal performance compared to even linear methods, mainly due to their inherent permutation equivariance that contradicts the causal nature of time series (Zeng et al., 2023c). Additionally, their quadratic time and memory complexity is a notable bottleneck hindering their use in large-scale long real-world settings with long-range prediction horizon.

In recent years, modern linear Recurrent Neural Networks (RNNs) have attracted much attention as the linear alternative to Transformers, improving Transformers' training and inference efficiency

while maintaining their effectiveness (Peng et al., 2023a; Katharopoulos et al., 2020; Kacham et al., 2023; Smith et al., 2023). While these models have shown promising performance on clean and tokenized data modalities such as language, applying them to multivariate time series modeling is more challenging as: (1) Contrary to text, time series data can be non-stationary and highly noisy, as demonstrated by complex temporal patterns. Accordingly, the additive nature of such recurrent models can cause error propagation in their predictions over time, requiring additional careful parametrization or design to achieve good performance (Jia et al., 2023; Behrouz et al., 2024d); (2) These models are inherently designed for a single sequence and so their use for time series data overlooks the importance of variate dependencies in modeling multivariate time series data (Zeng et al., 2023a; Zhang et al., 2023; Nie et al., 2023). Moreover, simply mixing the variates to take advantage of cross-variate information can hinder the performance in the general case as variate dependencies are not always useful in practice; e.g., when the target variate is not correlated with other covariates (Chen et al., 2023). Therefore, a major goal of effective modeling of multivariate time series is to develop a model which can *adaptively* mix cross variate information over time when appropriate; (3) To capture both cross-time and cross-variate information, several recent studies have sought to perform selective 2-dimensional recurrence across both variates (Jia et al., 2023; Behrouz et al., 2024d). These models, however, are sensitive to the order of variates, thus missing the permutation equivariance of information across variates.

**Contributions.** In our work, with the goal of mitigating the aforementioned limitations in existing time series models, we present LETO, a novel 2-dimensional architecture based on two meta in-context memory modules—called time and variate memory modules—that learns how to memorize cross-time and cross-variate patterns at test time, respectively. While LETO updates the time memory module using a recurrent rule to take advantage of its temporal inductive bias, it uses an attention-like (with `Softmax`) non-parametric memory module across variates to accurately consider their permutation equivariance property. To capture the discrete time dynamics of dependencies across variates, LETO needs to mix the states of both time and variate memories at each time stamps. However, the non-parametric nature of variate memory module makes it stateless, empowering the memory to learn the dynamics of variate dependencies across time. To overcome this challenge, LETO uses a parametric approximation of the non-parametric memory and expresses the `Softmax` attention using its Taylor series. To the best of our knowledge, LETO is the first native 2-dimensional hybrid model. In our experiments, we perform various evaluations and compare LETO with state-of-the-art time series models on diverse downstream tasks, including: (1) short-, long-, and ultra-long-term forecasting, (2) classification, and (3) anomaly detection tasks. We further demonstrate the effectiveness of LETO for longer horizons and support the significance of LETO's design by performing ablation studies.

## 2 PRELIMINARIES, BACKGROUND, AND RELATED WORK

In this section, we first discuss the notation that we use through the paper and then provide an overview of the background concepts and related studies. A more detailed discussions of the related work is in Appendix (B). Additionally, our model architecture is motivated by the following key directions: (1) meta learning, (2) learning to memorize, and (3) Titans Behrouz et al. (2024e). We provide a more detailed explanation of each of these topics in Appendix (A).

**Notation.** We let matrix $\mathbf{X} = \{\mathbf{x}_1, \ldots, \mathbf{x}_V\} \in \mathbb{R}^{V \times T \times d_{\text{in}}}$ denote a multivariate time series, where $T$ and $V$ are the number of time stamps and variates, respectively, and $d_{\text{in}}$ is the feature dimension of the input (often $d_{\text{in}} = 1$). We use $x_{v,t} \in \mathbb{R}^{d_{\text{in}}}$ to refer to the value of the time series in $v$-th variate at time $t$. In this paper, we mainly focus on forecasting, classification, and anomaly detection. In forecasting, given the historical series $\mathbf{X} = \{\mathbf{x}_1, \ldots, \mathbf{x}_V\}$, the model aims to predict the next $H$ time steps. For classification and anomaly detection, the task is to assign a label to the sequence, where anomaly detection is treated as a binary classification problem, labeling variate as "normal" or "anomaly".

**Autoregressive Process.** Autoregressive (AR) process is a basic but fundamental concept for time series modeling. An AR process models the causal nature of time series by writing each element as the linear combination of its past samples. Given $p \in \mathbb{N}$, $\mathbf{x}_k \in \mathbb{R}^d$, the linear autoregressive relationships between $\mathbf{x}_k$ and its past samples $\mathbf{x}_{k-1}, \mathbf{x}_{k-2}, \ldots, \mathbf{x}_{k-p}$ is modeled as

$$\mathbf{x}_k = \zeta_1 \mathbf{x}_{k-1} + \zeta_2 \mathbf{x}_{k-2} + \ldots, \zeta_p \mathbf{x}_{k-p} \qquad (\text{AR}(p) \text{ Process})$$

where $\zeta_1, \ldots, \zeta_p$ are coefficients. Note that we can simply extend the above autoregressive formulation to the multivariate setting by letting coefficients be vectors, replacing the product with element-wise product.

**Time Series Models.** The complexity of time series data—characterized by higher-order structures, multivariate dependencies, and domain variability—presents key challenges for model development. Models must capture both local and long-range dependencies, selectively leverage relevant covariates, and scale efficiently to long sequences without relying heavily on domain-specific pre-processing. Classical statistical models, such as ARIMA (Anderson & Kendall, 1976) and STL (Cleveland et al., 1990), effectively address periodic and trend components but are fundamentally limited when it comes to modeling non-linear and complex dependencies.

Early efforts to enhance time series forecasting with deep learning adopted recurrent neural networks (RNNs) (Elman, 1990) and their variants, such as Long Short-Term Memory (LSTM) networks (Hochreiter & Schmidhuber, 1997b) and Gated Recurrent Units (GRUs) (Cho et al., 2014), owing to their natural suitability for sequential data. Subsequently, temporal convolutional networks (TCNs) (Bai et al., 2018; Wang et al., 2023; Wu et al., 2022a) were introduced, excelling at capturing local patterns through carefully designed receptive fields.

The introduction of Transformer-based models (Vaswani et al., 2017) marked a significant advancement, enabling more effective modeling of both short and long term dependencies with enhanced scalability and predictive performance across a wide range of time series tasks (Wen et al., 2022). Transformer-style architectures such as (Liu et al., 2024c; Zhou et al., 2022b; Shi et al., 2024) demonstrate the power of attention to capture local and global temporal patterns, often enriching them with frequency-domain representations, downsampling, or mixture-of-experts components for improved efficiency. Building on this trend, recent multivariate forecasters further refine these ideas via frequency decomposition, patch-specific spatio-temporal filtering, non-stationarity-aware modules, and chunk-wise spatial correlation modeling with KAN's and FFT techniques (Huang et al., 2025b; Hu et al., 2025; Ma et al., 2025; Liu et al., 2025a; Si et al., 2025; Huang et al., 2025a), while general forecasting models with unified representations and adaptive transfer mechanisms, extend these backbones to cross-dataset settings (Wang et al., 2025b). However, the quadratic complexity of standard Transformers still poses optimization and scalability challenges (Zhou et al., 2021; Wu et al., 2021; Zhou et al., 2022b; Liu et al., 2021), motivating patch-based and hierarchical designs (Zhang & Yan, 2023; Nie et al., 2023; Chen et al., 2025b). Meanwhile, multilayer perceptrons have remained popular for forecasting due to their simplicity and direct mapping capabilities (Ekambaram et al., 2023).

Beyond forecasting, specialized architectures have been developed for anomaly detection and related tasks, for example channel-aware models that exploit frequency patching to detect multivariate anomalies (Wu et al., 2025). Finally, multi-dimensional recurrent models have recently attracted attention (Behrouz et al., 2024d; Meskin et al., 2025; Jia et al., 2023). Although their multi-dimensional recurrence can capture cross-time and cross-variate interactions, their recurrent nature across variates makes them sensitive to the order of variables, so performance can degrade under simple permutations; moreover, efficient training requires careful algorithmic design to parallelize the recurrences. Our design supports permutation equivariance over variates and remains effective and straightforward to train. For more discussion on limitations of existing model architectures see Section B.

**Test Time Memorization and Time Series Modeling.** In recent years, there have been growing interest in understanding the underlying mechanisms of sequence models and unifying (a subset of) them through a single perspective (Sun et al., 2024; Behrouz et al., 2025; Schlag et al., 2021; Liu et al., 2024a). In this work, we discuss a connection between test time memorization models, time series modeling, and autoregressive processes. In the associative memory perspective of sequence models, given the incoming input data $\mathbf{x}_t$, a sequence model is defined as an associative memory, $\mathcal{M}(\cdot)$ that aims to learn a mapping between a set of keys (i.e., $\{\boldsymbol{k}_i\}_{i=1}^N$) and values (i.e., $\{\boldsymbol{v}_i\}_{i=1}^N$) based on an objective function $\ell(\mathcal{M}(\cdot); \boldsymbol{k}_t, \boldsymbol{v}_t)$. For example, in recurrent neural networks, this memory module $\mathcal{M}$ is their hidden state. Since this memory module is updated for each incoming data (at test time), it is often called a test time learner or test time memorizer. It is notable that the process of training such memory is a meta learning process (Hospedales et al., 2021), where inside the inner-loop the corresponding parameters to memory are optimized, while the outer-loop optimizes other parameters in the neural network. For additional discussions on the meta learning process and how architectures

like Transformers and recurrent models can be formulated as associative memory module, we refer the reader to Behrouz et al. (2025) and our background discussion in Appendix (A).

In practice, given input data $\mathbf{x}_t$, keys and values are defined as the linear projections of the input, i.e.,

$$\boldsymbol{k}_t = W_k\mathbf{x}_t \qquad \text{and} \qquad \boldsymbol{v}_t = W_v\mathbf{x}_t, \tag{1}$$

where $W_k \in \mathbb{R}^d$ Another interpretation of this framework for associative memory is to view $\boldsymbol{k}_t$ as the corrupted version of the input, and define $\mathcal{M}(.)$ as a model that can reconstruct a projection of the input from the corrupted version. In this interpretation, objective $\ell(\mathcal{M}(\cdot); \boldsymbol{k}_t, \boldsymbol{v}_t)$ measures the ability of $\mathcal{M}$ in reconstructing the input projection. Despite the equivalence of these two interpretations, the latter provides an interesting connection between modeling time series data with sequence models. That is, modeling time series given a lookback window of $p$ time stamps in which the model aims to predict the next $h \geq 1$ steps, is equivalent to reconstructing a time series of $h + p$ time stamps from its corrupted version that masks its last $h$ steps. This reconstruction perspective and its connection to sequence models allow for the design of sequence models that are theoretically expressive and capable of modeling time series data. Despite this advantage, it is important to note that this formulation is limited to a single sequence. Hence, this begs the question: *"How can we design a native 2-dimensional model that learns to map underlying patterns of 2D data?"*

## 3 LETO: LEARNING TO MEMORIZE AT TEST TIME WITH 2-DIMENSIONAL MEMORY

To address this question, we present our model: LETO, a native 2-dimensional architecture that takes advantage of two separate memory modules, each of which learns how to memorize patterns across either time or variate dimensions.

### 3.1 HOW TO MEMORIZE 2-DIMENSIONAL DATA?

As discussed earlier, while sequence modeling and its test time memorization perspective can be an effective paradigm for modeling time series data, its design is limited to single sequences. Thus, for 2-dimensional data like multivariate time series, two memory modules are needed, each of which *learns* how to memorize patterns across each dimension (either time or variate) at test time. However, having memory modules that simply memorize the training data can significantly hinder the performance of the model, due to overfitting and the property that time series data at test time can be out-of-distribution (OOD). To this end, we utilize a meta in-context memory, where the model learns *how to memorize patterns at test time*. This memory does not directly memorize training data, but instead employs the underlying patterns in the training data to learn *what patterns* need to be memorized and what patterns need to be forgotten.

**Cross Time Dynamic.** For the sake of simplicity and to demonstrate the process of modeling cross-time patterns, we fix the variate to $v$ and remove it from subscript whenever the context is clear. Accordingly, for the input sequence this is a meta learning problem on the memory parameters, in which memory aims to reconstruct the projection of the time series (i.e., $\boldsymbol{v}_i = W_v\mathbf{x}_i$) from its corrupted version (i.e., $\boldsymbol{k}_i = W_k\mathbf{x}_i$). That is, given an internal objective $\ell(\cdot)$ that measures the quality of reconstruction, during the training process, the model performs two loops:

1. *Inner Loop*: In this loop the memory is optimized to reconstruct the sequence from its corrupted version using an optimization algorithm such as gradient descent. Therefore, the memory update is defined as:

$$\mathcal{M}_t = \alpha_t\mathcal{M}_{t-1} - \eta_t\nabla\ell(\mathcal{M}_{t-1}; \mathbf{x}_{v,t}), \tag{2}$$

   Note that in the inner loop we only optimize the memory parameters; other parameters are fixed in this loop.

2. *Outer Loop*: The outer loop is responsible for the training of the entire model for a specific downstream task such as forecasting, classification, or anomaly detection. In this process, while all parameters in the model are optimized, memory parameters are fixed.

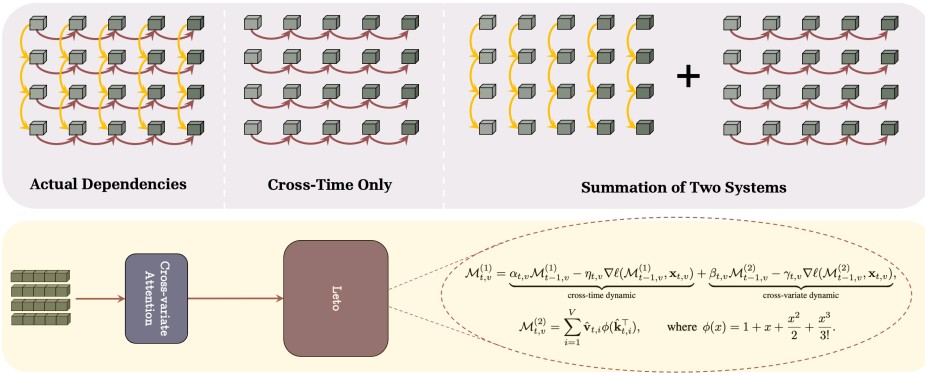

Figure 1: **An Overview of LETO's Architecture**: We define two inter-connected memory blocks $\mathcal{M}^1$, $\mathcal{M}^2$ corresponding to time and variate axes, where the recurrence is updated by fusing together both cross-time and cross-variate information, using an approximation of softmax attention for $\mathcal{M}^2$.

Using a reconstruction loss, i.e., $\ell(\mathcal{M}; \mathbf{x}_t) = \|\mathcal{M}\boldsymbol{k}_t - \boldsymbol{v}_t\|_2^2$, where $\boldsymbol{k}_t$ and $\boldsymbol{v}_t$ are defined as equation 1, gives us a memory module with delta update rule (recurrence) (Schlag et al., 2021) as:

$$\mathcal{M}_t = \mathcal{M}_{t-1} - \eta_t \nabla \ell(\mathcal{M}_{t-1}; \mathbf{x}_t) = (\mathbf{I} - \eta_t \boldsymbol{k}_t \boldsymbol{k}_t^\top)\mathcal{M}_{t-1} + \eta_t \boldsymbol{v}_t \boldsymbol{k}_t^\top, \quad (3)$$

where $(\mathbf{I} - \boldsymbol{k}_t \boldsymbol{k}_t^\top)$ is the transition matrix from state $\mathcal{M}_{t-1}$ to $\mathcal{M}_t$ and $\boldsymbol{v}_t \boldsymbol{k}_t^\top$ is the transformation of the input data. This linear recurrent process is equivalent to a linear dynamical system with non-diagonal transition matrix, which is more expressive than its counterpart dynamical systems with diagonal transition (Behrouz et al., 2024d; Patro & Agneeswaran, 2024; Li et al., 2024). In our later design of LETO in equation Variant 2, we further enhance the above formulation by incorporating a gating mechanism inspired by the Titans architecture (Behrouz et al., 2024e). Therefore, the update rule can be written as:

$$\mathcal{M}_t = (\alpha_t \mathbf{I} - \eta_t \boldsymbol{k}_t \boldsymbol{k}_t^\top)\mathcal{M}_{t-1} + \eta_t \boldsymbol{v}_t \boldsymbol{k}_t^\top, \quad (4)$$

where $\alpha$ controls the retention from the previous state of the memory. When $\alpha \to 1$, it fully retains the past state (equivalent to equation 3) and when $\alpha \to 0$ it erases the past state of the memory.

**Cross Variate Dynamic.** In the prior section, we discuss a neural memory module that learns how to memorize cross-time patterns. However, in multivariate time series data, the dependencies of variates can be a rich source of information, sometimes even more important than cross-time patterns (Tang et al., 2023; Behrouz et al., 2024a; Liu et al., 2024c). To this end, we aim to design a memory module that can learn from and memorize cross-variate patterns. One simple approach is to transpose the input data (re-ordering time and variate dimension) and apply our memory module introduced in equation 4 across variates. However, the main drawback of such a method is its sensitivity to the order of variates. That is, while the temporal inductive bias of recurrent models is effective for capturing temporal patterns, it is indeed a caveat that when sampling data, the order of elements are arbitrary. In multivariate time series data, the order of variates is often arbitrary and so we expect the model to produce the same output (or its corresponding permutation) when we change the order of variates. This property is called "permutation equivariance" (resp. "permutation invariant"), where the output of the model permutes the same (resp. remains the same) with the permutation of the input.

Transformers are one of the most powerful architectures with the permutation equivariance property (Yun et al., 2020; Xu et al., 2024). Although this property makes their direct applicability to time series data limited, it makes them a great choice of architectural backbone for use in learning the cross-variate information (Liu et al., 2024c). To this end, given the input data $\mathbf{X} = \{\mathbf{x}_1, \ldots, \mathbf{x}_V\} \in \mathbb{R}^{V \times T \times d_{\text{in}}}$, one can define $\tilde{\mathbf{X}} = \mathbf{X}^\top = \{\tilde{\mathbf{x}}_1, \ldots \tilde{\mathbf{x}}_T\} \in \mathbb{R}^{T \times V \times d_{\text{in}}}$ and then pass it to a Transformer block to capture the cross-variate dependencies:

$$\mathbf{Y} = \texttt{Transformer}\left(\tilde{\mathbf{X}}\right). \quad (5)$$

While the above method can satisfy both (1) fusing information across variates, and (2) preserving the robustness to the permutation of variates, it only models cross-variate patterns and misses the dynamics of variates dependencies (Behrouz et al., 2024d; Jia et al., 2023).

## 3.2 LETO: A NATIVE 2-DIMENSIONAL MEMORY SYSTEM

Previously we discussed how it is possible to design an effective memory module that learns how to map underlying patterns across time *or* variate dimensions in the data. A simple and commonly used method in the literature is to use two different modules, each for one of the dimensions, and then mix their outputs for the final prediction (Ahamed & Cheng, 2024b; Christou et al., 2024). That is, given input $\mathbf{X} \in \mathbb{R}^{V \times T \times d_{\text{in}}}$, one can use $\text{Module}_1(\cdot)$ and $\text{Module}_2(\cdot)$ to fuse information across time and variates, respectively, and then combine them for the final output:

$$Y_{\text{time}} = \text{Module}_1(\mathbf{X}), \qquad Y_{\text{variate}} = \text{Module}_2(\tilde{\mathbf{X}}),$$
$$Y_{\text{output}} = \text{Combine}\left(Y_{\text{time}}, Y_{\text{variate}}\right). \tag{Variant 1}$$

Another commonly used approach is to employ $\text{Module}_1(\cdot)$ and $\text{Module}_2(\cdot)$ in a sequential manner (instead of the above parallel manner). However, all these models treat each dimension separately and thus miss the inter-dependencies of time and variate dimensions at each state of the system, resulting in less expressive power in modeling time series data (see Theorem (1) for the details). To this end, we present a native 2-dimensional memory system that not only has the temporal inductive bias across time, but also has the permutation equivariance property across variates.

We use two memory modules $\mathcal{M}^{(1)}(\cdot)$ and $\mathcal{M}^{(2)}(\cdot)$ to learn the underlying mappings/patterns across time and variate dimensions, respectively. As discussed in section 2 and section 3, to design such memory modules it is appropriate to use a reconstruction objective $\ell(\cdot)$ for the memory and then optimize this objective with an optimization algorithm (such as gradient descent). However, to capture the inter-dependencies of dimensions at each step of optimization, it is necessary to fuse the information between the memory modules as well. Therefore, the state of each memory module not only depends on its time stamp, but it also depends on its variate. Given $\mathbf{X} = \{\mathbf{x}_1, \ldots, \mathbf{x}_V\}$ as the input, and arbitrary $v \in \{1, \ldots, V\}$ we define the update of cross-time memory, as:

$$\mathcal{M}_{t,v}^{(1)} = \alpha_{t,v}\mathcal{M}_{t-1,v}^{(1)} - \eta_{t,v}\nabla\ell(\mathcal{M}_{t-1,v}^{(1)}, \mathbf{x}_{t,v})$$
$$+ \beta_{t,v}\mathcal{M}_{t-1,v}^{(2)} - \gamma_{t,v}\nabla\ell(\mathcal{M}_{t-1,v}^{(2)}, \mathbf{x}_{t,v}), \tag{6}$$

where $\ell(\mathcal{M}_{t-1,v}^{(j)}, \mathbf{x}_{t,v}) = \|\mathcal{M}_{t-1,v}^{(j)}\boldsymbol{k}_{t.v} - \boldsymbol{v}_{t,v}\|_2^2$ for $j \in \{1, 2\}$ and $v \in \{1, \ldots, V\}$ and:

$$\boldsymbol{k}_{t,v} = W_k\mathbf{x}_{t,v}, \qquad \text{and} \qquad \boldsymbol{v}_{t,v} = W_v\mathbf{x}_{t,v}. \tag{7}$$

Expanding the gradient for the above formulation results in the recurrent update rule for the cross-time memory module as follows:

$$\mathcal{M}_{t,v}^{(1)} = (\alpha_{t,v}\mathbf{I} - \eta_{t,v}\boldsymbol{k}_{t,v}\boldsymbol{k}_{t,v}^\top)\mathcal{M}_{t-1,v} + \eta_{t,v}\boldsymbol{v}_{t,v}\boldsymbol{k}_{t,v}^\top$$
$$+ (\beta_{t,v}\mathbf{I} - \gamma_{t,v}\boldsymbol{k}_{t,v}\boldsymbol{k}_{t,v}^\top)\mathcal{M}_{t-1,v} + \gamma_{t,v}\boldsymbol{v}_{t,v}\boldsymbol{k}_{t,v}^\top. \tag{8}$$

The above formulation demonstrates how to update the cross-time memory. To get the final output from this memory, we need to multiply it by the input data $\mathbf{x}_{t,v}$ to achieve the $\mathbf{x}_{t,v}$'s corresponding information in the memory: i.e., $\mathbf{Y}_{t,v}^{(1)} = \mathcal{M}_{t,v}^{(1)}\mathbf{x}_{t,v}$. One can similarly define the recurrence for the cross-variate memory module $\mathcal{M}_{t,v}^{(2)}$ as:

$$\mathcal{M}_{t,v}^{(2)} = \theta_{t,v}\mathcal{M}_{t,v-1}^{(1)} - \lambda_{t,v}\nabla\ell(\mathcal{M}_{t,v-1}^{(1)}, \mathbf{x}_{t,v})$$
$$+ \mu_{t,v}\mathcal{M}_{t,v-1}^{(2)} - \omega_{t,v}\nabla\ell(\mathcal{M}_{t,v-1}^{(2)}, \mathbf{x}_{t,v}). \tag{9}$$

However, it is still sensitive to the order of variates. This sensitivity to variate ordering comes from the parametric nature of gradient descent algorithm as its iterations requires a series of ordered steps. Therefore, the use of any other parametric optimizer can cause such sensitivity to the order. To overcome this issue, we use the non-parametric estimate of our objective. Interestingly, with a small modification and usage of Nadaraya-Watson estimators (Fan, 2018; Zhang et al., 2022b), the non-parametric estimate of the objective is equivalent to softmax attention mechanism in Transformers (Vaswani et al., 2017), as also discussed in previous studies (Sun et al., 2024; Behrouz et al., 2025). As a result of this theoretical connection, we utilize an attention module for the cross-variate information mixing. The final output of this block can simply be defined as:

$$\mathbf{Y}_{t,v}^{(2)} = \theta_{t,v} \text{Attention}\left(\{\mathcal{M}_{t,i}^{(1)}\mathbf{x}_{t,i}\}_{i=1}^V\right)$$
$$+ \mu_{t,v} \text{Attention}\left(\{\mathbf{x}_{t,i}\}_{i=1}^V\right). \tag{10}$$

Table 1: Average performance on Ultra long-term forecasting tasks (MSE / MAE)

| Dataset | Metric | LETO | | MICN | | TimesNet | | PatchTST | | DLinear | | FiLM | | FEDformer | | Autoformer | | Informer | |
|---|---|---|---|---|---|---|---|---|---|---|---|---|---|---|---|---|---|---|---|
| | | MSE | MAE | MSE | MAE | MSE | MAE | MSE | MAE | MSE | MAE | MSE | MAE | MSE | MAE | MSE | MAE | MSE | MAE |
| ECL | 720–1440 | 0.4782 | 0.5614 | 1.0460 | 0.7765 | 0.6119 | 0.5962 | 0.8243 | 0.6704 | 0.4923 | 0.5473 | 0.4730 | 0.5336 | 0.4833 | 0.5393 | 1.4957 | 0.9533 | 0.5064 | 0.5317 |
| | 1440–1440 | 0.4639 | 0.5387 | 0.8262 | 1.2207 | 0.5720 | 0.5712 | 0.9053 | 0.7328 | 0.5146 | 0.5615 | 0.4849 | 0.5429 | 0.5142 | 0.5571 | 1.7873 | 1.0283 | 0.7247 | 0.6920 |
| | 1440–2880 | 0.6047 | 0.5868 | 2.8936 | 1.3717 | 0.7683 | 0.6846 | 1.1282 | 0.8087 | 0.8355 | 0.7193 | 0.6847 | 0.6493 | 3.9018 | 1.5276 | 1.2867 | 0.8878 | 0.6152 | 0.5953 |
| Traffic | 720–1440 | 0.1672 | 0.2431 | 0.2876 | 0.3916 | 0.1882 | 0.2656 | 0.1904 | 0.2685 | 0.1639 | 0.2412 | 0.1638 | 0.2448 | 0.2753 | 0.3650 | 0.3104 | 0.4095 | 0.7614 | 0.6496 |
| | 1440–1440 | 0.1521 | 0.2497 | 0.2905 | 0.3923 | 0.2081 | 0.2712 | 0.1917 | 0.2764 | 0.1590 | 0.2411 | 0.1602 | 0.2437 | 0.2848 | 0.3681 | 0.2970 | 0.3999 | 0.7375 | 0.6414 |
| | 1440–2880 | 0.1425 | 0.2433 | 0.2823 | 0.3874 | 0.1560 | 0.2409 | 0.1819 | 0.2761 | 0.1550 | 0.2421 | 0.1744 | 0.2693 | 0.2952 | 0.3844 | 0.3035 | 0.3982 | 0.9408 | 0.7618 |
| ETTh1 | 720–1440 | 0.1331 | 0.2943 | 0.4640 | 0.5836 | 0.1391 | 0.3049 | 0.3708 | 0.4906 | 0.2952 | 0.4370 | 0.2949 | 0.4388 | 0.1768 | 0.3409 | 0.3298 | 0.4741 | 0.1378 | 0.3051 |
| | 1440–1440 | 0.1359 | 0.3120 | 0.5188 | 0.6075 | 0.1404 | 0.3093 | 0.4475 | 0.5392 | 0.2200 | 0.3714 | 0.3226 | 0.4678 | 0.1928 | 0.3576 | 0.3618 | 0.5507 | 0.1402 | 0.3192 |
| | 1440–2880 | 0.2591 | 0.3949 | 0.7591 | 0.7215 | 0.2732 | 0.4094 | 0.9617 | 0.8072 | 0.3773 | 0.4794 | 0.3624 | 0.4705 | 0.2627 | 0.3754 | 0.3177 | 0.4733 | 0.3495 | 0.4111 |

Table 2: Average performance on short-term forecasting tasks on the M4 dataset. A lower SMAPE, MASE, and OWA indicate better prediction. * is an abbreviation of the "former" term.

| Models | LETO (Ours) | ROSE 2025 | ModernTCN 2024 | TimeMixer 2024 | PatchTST 2023 | TimesNet 2023 | N-HiTS 2022 | N-BEATS 2019 | ETS* 2022 | LightTS 2022 | DLinear 2023 | FED* 2022 | Stationary 2022 | Auto* 2021 |
|---|---|---|---|---|---|---|---|---|---|---|---|---|---|---|
| SMAPE | 11.658 | 11.764 | 11.698 | 11.723 | 11.807 | 11.829 | 11.927 | 11.851 | 14.718 | 13.525 | 13.639 | 12.840 | 12.780 | 12.909 |
| MASE | 1.541 | 1.568 | 1.556 | 1.559 | 1.590 | 1.585 | 1.613 | 1.599 | 2.408 | 2.111 | 2.095 | 1.701 | 1.756 | 1.771 |
| OWA | 0.832 | 0.871 | 0.838 | 0.840 | 0.851 | 0.851 | 0.861 | 0.855 | 1.172 | 1.051 | 1.051 | 0.918 | 0.930 | 0.939 |

(Weighted Average)

Note that $\mathcal{M}_{t,i}^{(1)} \mathbf{x}_{t,i}$ provides the $\mathbf{x}_{t,i}$'s corresponding information in cross-time memory module and so the first term combines the cross-time dynamic of all variates at the same time. While computation of the final output for the cross-variate memory is clear, we need to access its memory (i.e., $\mathcal{M}_{t,v}^{(2)}$) to use in the update of cross-time memory (i.e., equation 6). The memory of Transformers are known to be the pair of key and value matrices $(\mathbf{K}, \mathbf{V})$ in the attention mechanism (Zhang & Cai, 2022; Wu et al., 2022c; Behrouz et al., 2024e; Bietti et al., 2023). However, incorporating a pair of matrices into the recurrence update rule of equation 6 is unclear and challenging. Therefore, we utilize a kernelized variant of attention, in which we replace `Softmax` with a separable kernel $\phi(\cdot)$ (Katharopoulos et al., 2020; Kacham et al., 2023; Arora et al., 2024) (see Appendix (A) for the corresponding background and detailed formulation). This allows us to concretely define the memory of the Transformer with keys and values of $\{\hat{\boldsymbol{k}}_i\}$ and $\{\hat{\boldsymbol{v}}_i\}$ as (Katharopoulos et al., 2020):

$$\mathcal{M}_{t,v}^{(2)} = \sum_{i=1}^{V} \hat{\boldsymbol{v}}_{t,i} \phi(\hat{\boldsymbol{k}}_{t,i}^{\top}). \qquad (11)$$

The question regarding what would be the optimal kernel $\phi(\cdot)$ to use in the above formulation remains. To answer this, we recall the formulation of `Softmax` attention that is proportional to `softmax`$(\boldsymbol{q}_t^{\top} \boldsymbol{k}_t)\boldsymbol{v}_t$. To replace softmax `softmax`$(\cdot)$ with a separable kernel $\phi(\cdot)$, we can choose the kernel to approximate the exponential term in softmax with its Taylor series. Accordingly, we use the first four terms of the Taylor series approximation of the exponential function: $\exp(\cdot)$ defined as:

$$\exp(x) \approx \phi(x) = 1 + x + \frac{x^2}{2} + \frac{x^3}{3!}. \qquad (12)$$

Combining the prior expressions, we can define our native 2-dimensional update rule as:

$$\begin{aligned}
\mathcal{M}_{t,v}^{(1)} = &\ \alpha_{t,v}\mathcal{M}_{t-1,v}^{(1)} - \eta_{t,v}\nabla\ell(\mathcal{M}_{t-1,v}^{(1)}, \mathbf{x}_{t,v}) \\
&+ \beta_{t,v}\mathcal{M}_{t-1,v}^{(2)} - \gamma_{t,v}\nabla\ell(\mathcal{M}_{t-1,v}^{(2)}, \mathbf{x}_{t,v}),
\end{aligned} \qquad \text{(Variant 2)}$$

where $\mathcal{M}_{t,v}^{(2)} = \sum_{i=1}^{V} \hat{\boldsymbol{v}}_{t,i}\phi(\hat{\boldsymbol{k}}_{t,i}^{\top})$ and $\phi(x) = x + \frac{x^2}{2} + \frac{x^3}{3!}$. Note that in the above formulation $\hat{\boldsymbol{v}}_i$ and $\hat{\boldsymbol{k}}_i$ are keys and values of the Transformer block, coming from the keys and values of the cross-variate dynamic attention mentioned in equation 10. In the following theorem, applicable to the linear recurrence variant, we demonstrate that this inter-connectivity of these two memories can enhance the expressive power of model, compared to utilizing two separate memory modules:

**Theorem 1.** *Let* `Module`$_i(\cdot)$ *be linear recurrent models, then inter-connected memory modules (i.e., equation Variant 2) can express full-rank kernels with $\mathcal{O}(1)$ parameters, while independent memory systems (i.e., equation Variant 1) require at least $\mathcal{O}(N)$ parameters to express matrix with rank $N$.*

Table 3: Average performance on long term forecasting tasks over four prediction lengths: {96, 192, 336, 720}. A lower MAE and MSE indicates a better prediction. As a convention for all experimental results, the best performance is highlighted in **red**, and the second-best is underlined.

| Models | LETO (Ours) | | TimePro | | TimeFilter | | TimeKAN | | TimeMixer | | Simba | | ModernTCN | | iTransformer | | RLinear | | PatchTST | | Crossformer | | TiDE | | TimesNet | | DLinear | |
|---|---|---|---|---|---|---|---|---|---|---|---|---|---|---|---|---|---|---|---|---|---|---|---|---|---|---|---|---|
| | MSE | MAE | MSE | MAE | MSE | MAE | MSE | MAE | MSE | MAE | MSE | MAE | MSE | MAE | MSE | MAE | MSE | MAE | MSE | MAE | MSE | MAE | MSE | MAE | MSE | MAE | MSE | MAE |
| ETTm1 | 0.347 | 0.375 | 0.391 | 0.400 | 0.377 | 0.393 | 0.376 | 0.395 | 0.381 | 0.385 | 0.383 | 0.396 | 0.351 | 0.381 | 0.407 | 0.410 | 0.414 | 0.407 | 0.387 | 0.400 | 0.513 | 0.496 | 0.419 | 0.419 | 0.400 | 0.406 | 0.403 | 0.407 |
| ETTm2 | 0.249 | 0.302 | 0.281 | 0.326 | 0.272 | 0.321 | 0.277 | 0.322 | 0.275 | 0.323 | 0.271 | 0.327 | 0.253 | 0.314 | 0.288 | 0.332 | 0.286 | 0.327 | 0.281 | 0.326 | 0.757 | 0.610 | 0.358 | 0.404 | 0.291 | 0.333 | 0.350 | 0.401 |
| ETTh1 | 0.393 | 0.401 | 0.438 | 0.438 | 0.420 | 0.428 | 0.417 | 0.427 | 0.447 | 0.440 | 0.441 | 0.432 | 0.404 | 0.420 | 0.454 | 0.447 | 0.446 | 0.434 | 0.469 | 0.454 | 0.529 | 0.522 | 0.541 | 0.507 | 0.458 | 0.450 | 0.456 | 0.452 |
| ETTh2 | 0.318 | 0.381 | 0.377 | 0.403 | 0.364 | 0.397 | 0.383 | 0.404 | 0.364 | 0.395 | 0.361 | 0.391 | 0.322 | 0.379 | 0.383 | 0.407 | 0.374 | 0.398 | 0.387 | 0.407 | 0.942 | 0.684 | 0.611 | 0.550 | 0.414 | 0.427 | 0.559 | 0.515 |
| Exchange | 0.297 | 0.364 | 0.352 | 0.399 | 0.324 | 0.383 | 0.330 | 0.389 | 0.391 | 0.453 | 0.298 | 0.363 | 0.302 | 0.366 | 0.360 | 0.403 | 0.378 | 0.417 | 0.367 | 0.404 | 0.940 | 0.707 | 0.370 | 0.413 | 0.416 | 0.443 | 0.354 | 0.414 |
| Traffic | 0.408 | 0.267 | 0.430 | 0.291 | 0.409 | 0.270 | 0.407 | 0.268 | 0.484 | 0.297 | 0.493 | 0.291 | 0.398 | 0.270 | 0.428 | 0.282 | 0.626 | 0.378 | 0.481 | 0.304 | 0.550 | 0.304 | 0.760 | 0.473 | 0.620 | 0.336 | 0.625 | 0.383 |
| Weather | 0.216 | 0.253 | 0.251 | 0.276 | 0.239 | 0.269 | 0.242 | 0.272 | 0.240 | 0.271 | 0.255 | 0.280 | 0.224 | 0.264 | 0.258 | 0.278 | 0.272 | 0.291 | 0.259 | 0.281 | 0.259 | 0.315 | 0.271 | 0.320 | 0.259 | 0.287 | 0.265 | 0.317 |
| ECL | 0.149 | 0.247 | 0.169 | 0.262 | 0.158 | 0.256 | 0.197 | 0.286 | 0.182 | 0.272 | 0.185 | 0.274 | 0.156 | 0.253 | 0.178 | 0.270 | 0.219 | 0.298 | 0.205 | 0.290 | 0.244 | 0.334 | 0.251 | 0.344 | 0.192 | 0.295 | 0.212 | 0.300 |

Table 4: Ablation Study of LETO on ETT, Weather, and Exchange datasets

| Model | ETTh1 MSE / MAE | ETTh2 MSE / MAE | ETTm1 MSE / MAE | ETTm2 MSE / MAE | Weather MSE / MAE | Exchange MSE / MAE |
|---|---|---|---|---|---|---|
| Full LETO | **0.393 / 0.401** | **0.318 / 0.381** | **0.347 / 0.375** | **0.243 / 0.302** | **0.216 / 0.253** | **0.297 / 0.364** |
| w/o Cross Variate Attention | 0.458 / 0.447 | 0.400 / 0.427 | 0.394 / 0.419 | 0.320 / 0.362 | 0.244 / 0.274 | 0.311 / 0.398 |
| w/o Linear Attention | 0.454 / 0.454 | 0.392 / 0.421 | 0.407 / 0.410 | 0.341 / 0.370 | 0.258 / 0.278 | 0.360 / 0.403 |
| w/o Weighted Gating | 0.405 / 0.412 | 0.368 / 0.392 | 0.389 / 0.397 | 0.312 / 0.354 | 0.237 / 0.269 | 0.301 / 0.384 |

## 3.3 MODEL DESIGN OF LETO

While our recurrence formulation is theoretically motivated to capture both cross-time and variate dependencies, its training can be difficult due its recurrent nature, potentially limiting parallelizable training. In this section, we discuss the architectural details in LETO and present a fast parallelizable training approach. We refer the reader to figure 1 for an illustration of the design of LETO.

**Parallelizable Training.** Despite the recurrent nature of LETO, in this section, we build upon the training algorithms of Sun et al. (2024) and Behrouz et al. (2024e) and present a parallel training process for our model. To begin, given a variate $v$, we divide its corresponding time series $\{\mathbf{x}_{1,v}, \ldots, \mathbf{x}_{T,v}\}$ with length $T$ into $C$ subsequences of length $b = \frac{T}{C}$, each of which is represented by $\mathcal{S}_i = \{\mathbf{x}_{(i-1)b+1,v}, \ldots, \mathbf{x}_{ib,v}\}$. Recall that the cross-variate dynamic term in equation 10 is independent of time and variate states in our formulation and thus can be computed in advance. Note that the training procedure for the attention module is parallelizable. Given the output of the attention module, we can also calculate all the states of $\mathcal{M}^{(2)}$ memory using equation 11. Therefore, we can calculate the gradient term with respect to $\mathcal{M}^{(2)}$ in (Variant 2), all in advance. Having the states of $\mathcal{M}^{(2)}$ and its corresponding gradient terms, we have calculated the cross-variate dynamic term in (Variant 2) in advance and so we only need to compute the cross-time dynamic term in a parallelizable manner. To this end, following the algorithms of Sun et al. (2024) and Behrouz et al. (2024e), we approximate the gradient term $\nabla \ell(\mathcal{M}^{(1)}_{t-1,v}, \mathbf{x}_{t,v})$ with $\nabla \ell(\mathcal{M}^{(1)}_{t',v}, \mathbf{x}_{t,v})$, in which $t'$ is the last state of the memory in the previous chunk, i.e., $t' = \lfloor \frac{t}{b} \rfloor \times b$. Therefore, we can calculate the gradients of each chunk in advance, making the recurrence linear, which is highly parallelizable. For a detailed discussion of parallelizable training, including pseudocode, see Appendix (C). Thus, we can parallelize the training process for each variate, and by scanning the variates from top to bottom, we can encode all the states in the multivariate time series. Note that the training complexity is linear across time and is dominated by the attention module's complexity across variates. Furthermore, in our experiments meta in-context memory states are reset per sequence and receptive fields are matched across baselines-specifically, no cross-batch or cross-sequence state is preserved.

## 4 EXPERIMENTS

**Goals and Baselines.** In this section, we evaluate LETO on a wide range of time series tasks, comparing with the most recent state-of-the-art multivariate time series models (Wu et al., 2023; Luo & Wang, 2024; Lim & Zohren, 2021; Woo et al., 2022; Wu et al., 2021; Zhou et al., 2022b; Zhang & Yan, 2023; Liu et al., 2024c; Dehghani et al., 2023; Das et al., 2023; Liu et al., 2022a; Patro & Agneeswaran, 2024; Zeng et al., 2023b; Xu et al., 2021; Wang et al., 2024; 2025b; Huang et al., 2025b; Ma et al., 2025; Hu et al., 2025) on forecasting: long, ultra-long, and short term, classification, and anomaly detection tasks. Next, we evaluate the significance of the LETO's components by performing ablation studies. Dataset descriptions, complete experimental results, visualization of

predictions, hyperparameters, metric descriptions, and full experimental results on the effect of other design choices are provided in D. We control the effect of parameters and all models use the same number of parameters and hyperparameters for training and evaluation. We did not use the "drop-last" operation Qiu et al. (2024) in our data loaders for any of our experiments. All batches, including the final, possibly smaller batches, were used in the training.

### 4.1 MAIN RESULTS: CLASSIFICATION AND FORECASTING

**Long-Term Forecasting.** We conduct experiments on the long-term forecasting tasks using commonly used benchmark datasets used by Zhou et al. (2021) and many others. The average performance across different horizons is summarized in Table 3. LETO consistently delivers competitive results across different datasets, highlighting its robustness compared to recurrent, convolutional, SSM, and Transformer-based models.

**Ultra Long-term Forecasting.** We further extend the evaluation to ultra-long-range forecasting on the same benchmark datasets (Zhou et al., 2021) to observe the effectiveness of LETO in longer horizons. The tasks on the left side of the Table 1 retain the same interpretation as in the standard long-term forecasting setting. The results in Table 1 demonstrate LETO's ability to capture long-term dependencies from extremely long historical inputs, maintaining its steady performance across various significantly extended prediction horizons.

**Classification and Anomaly Detection.** We evaluate the performance of LETO on 10 multivariate datasets from the UEA Time Series Classification Archive (Bagnall et al., 2018) (see figure 2 and Table 15). For anomaly detection, which is typically treated as a binary classification task, we conduct experiments on five widely-used benchmarks: SMD (Su et al., 2019), SWaT (Mathur & Tippenhauer, 2016), PSM (Abdulaal et al., 2021), and SMAP (Hundman et al., 2018) (see figure 2 and Table 14).

**Short-Term Forecasting.** Our evaluation on short-term forecasting tasks using the M4 benchmark datasets (Godahewa et al., 2021) is reported in Table 2 (with the full results provided in Table 11). We fix the input length to twice the prediction length and calculate Symmetric Mean Absolute Percentage Error (SMAPE), Mean Absolute Scaled Error (MASE), and Overall Weighted Average (OWA) as the evaluation metrics.

Figure 2: Anomaly detection and classification results of LETO and baselines. Higher accuracy/F1-score indicate better performance.

### 4.2 ABLATION STUDY

To validate the effectiveness of our model design, we conduct an ablation study on long-term forecasting tasks averaged over 5 runs on ETT, Weather, and Exchange datasets by removing key architectural components-see Table 4. The first row reports the LETO's performance, while the second row removes the cross attention block, the third row removes the the linear attention mechanisms, and the fourth row removes the the weights for the final gating between each block. The results demonstrate that LETO containing all components yields the strongest performance. Notably, the results without the linear attention component and Transformer block perform the worst, highlighting the importance of maintaining separate time and variate memories, and incorporating both in the recurrence in order to capture their interdependencies. A more extensive ablation, varying the Taylor expansion order, chunk size, and cross-memory coupling strength, is provided in Appendix E.2.

## 5 CONCLUSION

We present LETO, a native 2-dimensional memory module that takes the advantage of temporal inductive bias across time while maintaining the permutation equivariance of variates. LETO uses a meta in-context memory module to learn and memorize patterns across time dimension, and simultaneously, incorporates information from other correlated variates, if it is needed. Our experimental and theoretical results support the effectiveness of LETO across a diverse set of time series tasks.

## 6 REPRODUCIBILITY STATEMENT

We provide the relevant code for our model. All proofs are provided in the appendix with explanations and underlying assumptions. A complete description of the datasets used in our experiments are provided as well in the appendix.

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

## A  PRELIMINARIES AND BACKGROUND

**Transformers and their Permutation Equivariance Property.** Transformers (Vaswani et al., 2017) have been the de facto backbone for many deep learning models and are based on the attention module. Let $x \in \mathbb{R}^{N \times d_{\text{in}}}$ be the input, attention computes output $\mathbf{y} \in \mathbb{R}^{N \times d_{\text{in}}}$ based on softmax over input dependent key, value, and query matrices:

$$\mathbf{Q} = x\mathbf{W_Q}, \qquad \mathbf{K} = x\mathbf{W_K}, \qquad \mathbf{V} = x\mathbf{W_V}, \tag{13}$$

$$\mathbf{y}_i = \sum_{j=1}^{N} \frac{\exp\left(\mathbf{Q}_i^\top \mathbf{K}_j / \sqrt{d_{\text{in}}}\right) \mathbf{V}_j}{\sum_{\ell=1}^{N} \exp\left(\mathbf{Q}_i^\top \mathbf{K}_\ell / \sqrt{d_{\text{in}}}\right)}, \tag{14}$$

where $\mathbf{W_Q}, \mathbf{W_K}$, and $\mathbf{W_V} \in \mathbb{R}^{d_{\text{in}} \times d_{\text{in}}}$ are learnable parameters. This formulation of attention makes it permutation equivariant, meaning that the permutation of the input cannot change the output but permute it. That is, let $\pi(.)$ be a permutation, and $\mathcal{A}(\cdot)$ be the above attention module, we have:

$$\mathcal{A}(\pi(x)) = \pi(\mathcal{A}(x)). \tag{15}$$

The property, which is called permutation equivariance, is a desirable property for the data that is permutation equivariant, such as variates in the multivariate time series. When encoding the multivariate time series, we do not want the output of the model to be sensitive to the order of the input (variates) and so transformers are great architectures as any change to the order, does not change the output, but just permutes it.

**Learning to Memorize at Test Time.** The concept of learning to memorize at test time is derived from the learning at test time or learning to learn, which backs to very early studies on local learning Bottou & Vapnik (1992): i.e., training each test sample on its neighbors before making a prediction (Zhang et al., 2006; Gandelsman et al., 2022). Later, test time training shows promising results in vision tasks (Jain & Learned-Miller, 2011; Mullapudi et al., 2019), mainly because of the ability to properly address out-of-distribution cases. Using this perspective, recently this idea has been applied on sequence modeling (Sun et al., 2024; Behrouz et al., 2024e; 2025). These methods that aim to train a memory module that learns how to memorize the context at test time, have shown promising results in language and sequence modeling tasks. In this work, we also take this perspective and design a 2-dimensional test time memorizer that generalizes all these methods to 2-dimensional data modality.

## B  ADDITIONAL RELATED WORKS AND LIMITATIONS OF EXISTING FRAMEWORKS

**Classical Approach.** Time series modeling has been a fundamental research topic, Classical approaches include a range of statistical models such as exponential smoothing (Winters, 1960), ARIMA (Bartholomew, 1971), SARIMA (Bender & Simonovic, 1994), and the Box-Jenkins methodology (Box & Jenkins, 1968), with later advancements introducing state-space models (Harvey, 1990; Aoki, 2013). While these models offer interpretability, they often fall short in capturing complex non-linear dynamics and typically rely on manual inspection of time series characteristics—such as trend and seasonality—limiting their adaptability across diverse datasets.

**Transformer-based models.** Transformer-based architectures have become increasingly prominent in multivariate time series forecasting, particularly when modeling complex inter-variable and temporal dependencies (Zhou et al., 2022b; Kitaev et al., 2020; Zhang & Yan, 2023; Zeng et al., 2023a; Zhou et al., 2021; Liu et al., 2021; Wu et al., 2021; Ilbert et al., 2024; Nie et al., 2023). A line of research has focused on designing specialized attention mechanisms that leverage the unique structure of time series data (Woo et al., 2022), while others have explored strategies for capturing long-term temporal patterns to improve forecasting accuracy (Nie et al., 2023; Zhou et al., 2022a).

In parallel, recent works have revisited linear recurrent neural networks (Linear RNNs) as efficient alternatives to Transformers, aiming to reduce the quadratic complexity while maintaining competitive performance on long-range dependency modeling (Sun et al., 2023; Peng et al., 2023b; Wu et al., 2023). For instance, Chen et al. (2023) introduce TSMixer, a purely MLP-based model that

demonstrates strong performance on time series forecasting tasks. Notably, the expressive capacity of certain linear models aligns with 2D state space models (SSMs), suggesting that these architectures can be interpreted as specific instances within the broader 2D SSM framework. Additionally, convolution-based models have shown renewed promise (Luo & Wang, 2024), where the use of global convolutional kernels facilitates an expanded receptive field for capturing long-range dynamics.

More recently, several multivariate forecasters build on Transformer-style patching or hierarchical designs. TimeKAN (Huang et al., 2025b) introduces a KAN-based frequency decomposition over temporal patches; TimeFilter (Hu et al., 2025) constructs patch-specific spatio-temporal graphs and filters them to emphasize informative frequency components; and TimePro (Ma et al., 2025) proposes variable- and time-aware hyper-states to efficiently capture multivariate dynamics. These models can be viewed as sophisticated 1D (temporal) architectures augmented with channel-mixing modules. In contrast, LETO maintains two coupled meta-memories that are updated jointly along the temporal and variate axes, providing a native 2D view of multivariate dynamics rather than treating cross-variate interactions as a post-hoc mixer on top of a purely temporal backbone.

**Recurrent-based models.** Another line of research closely related to our work involves deep sequence modeling. Recurrent neural networks (RNNs), including variants such as GRUs (Chung et al., 2014), LSTMs (Hochreiter & Schmidhuber, 1997a), and DeepAR (Salinas et al., 2020), have been widely used for sequential data. However, these models suffer from well-known limitations such as vanishing and exploding gradients, along with inherently sequential computation that slows down training and inference. To address these inefficiencies, recent efforts have explored linear attention mechanisms as faster alternatives (Katharopoulos et al., 2020; Schlag et al., 2021; Kacham et al., 2023). For instance, Katharopoulos et al. (2020) propose a linear attention model with a recurrent formulation, enabling efficient inference and reduced computational complexity.

In parallel, deep state space models (SSMs) have gained momentum as a compelling alternative to Transformer-based architectures (Vaswani et al., 2017), offering improved scalability and training efficiency (Gu et al., 2020). These models blend classical state space formulations with deep learning by parameterizing neural network layers using multiple linear SSMs. This hybrid formulation leverages the convolutional interpretation of SSMs to mitigate the optimization challenges typically associated with RNNs (Smith et al., 2023). Recently, Gu & Dao (2023) introduced Mamba, a novel deep SSM architecture where parameters dynamically depend on input features. This SSM based approach has been successfully extended to various modalities—including images (Ma et al., 2024; Liu et al., 2024d; Behrouz et al., 2024c), point clouds (Liang et al., 2024), tabular data (Ahamed & Cheng, 2024a), graphs (Behrouz & Hashemi, 2024b; Behrouz et al., 2024b; Huang et al., 2024), and time series (Behrouz et al., 2024d; Cao et al., 2025; Ahamed & Cheng, 2024b; Patro & Agneeswaran, 2024)—demonstrating strong capabilities in modeling long-range dependencies across domains.

**Time-series foundation models.** A recent line of work aims to build time-series foundation models (TSFMs) that learn a single universal backbone from large, heterogeneous collections of time series and adapt it to many downstream tasks (forecasting, classification, anomaly detection, imputation, *etc.*). Representative examples include models that learn unified sequence representations and adaptive transfer mechanisms using FFT based techniques (Wang et al., 2025b; Benechehab et al., 2025), pattern machines such as TimeMixer++ that scale patch-based temporal mixers across domains (Wang et al., 2025a), and large TSFM families such as Sundial and Moirai-MoE that push capacity via dense or sparse mixture-of-experts architectures (Liu et al., 2025c;b). These works primarily study scaling laws, cross-dataset pretraining, and transfer protocols (zero-shot, few-shot, or light fine-tuning), typically building on standard Transformer- or MLP-style backbones.

Our focus is complementary and orthogonal to TSFMs. LETO is a native 2D meta-memory architecture designed for supervised, per-dataset forecasting at a fixed parameter budget, with no cross-dataset pretraining and a model size comparable to strong supervised baselines such as ModernTCN and iTransformer. Directly comparing our results to TSFMs would simultaneously conflate (i) architectural inductive bias, (ii) model scale, and (iii) pretraining data. Instead, we treat LETO as a building block at the same scale as existing supervised models: in principle, the LETO cell could be used as the backbone inside a TSFM pipeline (replacing a Transformer or MLP block) to provide an explicitly coupled temporal–variate memory within large pretrain-and-transfer frameworks. Exploring such integrations is an interesting direction for future work.

**Other Methods.** Graph-based models have emerged as powerful tools for time series forecasting(Wu et al., 2020; Yi et al., 2024), especially when the data exhibits spatial or relational structure across variables or entities. Approaches such as graph neural networks (GNNs) model dependencies through learned graph representations, enabling effective spatiotemporal forecasting in domains like traffic (Yu et al., 2017; Li et al., 2017) and sensor networks (Wu et al., 2019). Recent work has extended these ideas by incorporating dynamic graphs (Wu et al., 2023; Dwivedi et al., 2022; Gastinger et al., 2024), learning graph structures jointly with temporal dynamics to better capture evolving relationships over time. These methods offer strong performance in settings where explicit or latent graph structure underpins multivariate time series behavior.

**Limitations of Prior Frameworks.** Recent diagnostic and theoretical studies have made the limitations of existing time–series backbones much more concrete. A growing body of work shows that vanilla Transformer, MLP, and linear global models, when applied to multivariate time series data, often lack strong temporal inductive bias, under–utilize cross–series structure, or require fragile evaluation protocols to appear competitive. For example, Chen et al. (2025a) systematically compare point–wise, patch–wise, and variate–wise Transformers and find that performance on standard long–horizon benchmarks is dominated by intra–variate dependencies, with inter–variate attention contributing relatively little and success relying heavily on Z–score normalization and (approximate) stationarity of each series. Theoretical work on in–context forecasting further shows that simplified Transformer variants with linear self–attention cannot outperform classical linear predictors on AR($p$) processes and exhibit a provably non–vanishing performance gap at finite context length, despite quadratic complexity. Zhou et al. (2025) Recurrent and convolutional alternatives address temporal inductive bias but introduce other trade–offs: linear RNNs provide a natural causal prior yet are inherently single–sequence and prone to error accumulation, Meskin et al. (2025) while ModernTCN–style architectures require careful handling of data loading, validation, and "drop–last" to avoid overly optimistic conclusions.Akacik & Hoogendoorn (2025) Complementary work on context neural networks and related global models emphasizes that many "global univariate" approaches still forecast each series in isolation at inference time, leaving useful cross–series context under–exploited unless one is willing to pay the cost of full attention or graph–based modeling. Sriramulu et al. (2024). At the multivariate architectural level, several recent papers explicitly document limitations of current ways of mixing temporal and variate information. UniTST Liu et al. (2024b) shows that many multivariate time series Transformers—including iTransformer Liu et al. (2024c), and Crossformer Zhang & Yan (2023) apply time–wise and variate–wise attention in separate stages (sequentially or in parallel) and therefore cannot directly model cross–time *and* cross–variate dependencies in a single operation; they demonstrate that such cross–time cross–variate links are present and beneficial in real–world data. Independent analyses of TimesNet and its variants report that embedding all variates at a single time step into one token can blur physically different signals and degrade multivariate performance, motivating "inverted" designs that are more variate–centric. Hu & Li (2024). HYDRA Meskin et al. (2025) formalizes a related gap: architectures that only mix along one dimension at a time (e.g., separate temporal and channel mixers, or pure variate–wise attention as in iTransformer) implement a restricted class of kernels, while a genuine 2D recurrence that updates a time memory and a variate memory jointly at each step can represent strictly higher–rank interactions between time and variates. LETO is designed precisely in response to these documented limitations: it combines a contractive recurrent temporal memory that encodes a strong causal inductive bias with a permutation–equivariant cross–variate memory, and an explicit coupling between the two memories at every time step. Our kernel analysis and ablations show that this hybrid 2D meta–memory strictly enlarges the class of realizable kernels compared to pure variate–wise attention, thereby providing an analytical separation from iTransformer–style architectures while remaining at the same parameter scale. An extended discussion of limitations of various time series architecture is beyond the scope of this work.

## C    PARALLELIZABLE TRAINING OF LETO

While the recurrence-based formulation of LETO enables it to better capture joint temporal and variate dependencies, as well as their independent dynamics, it introduces sequential dependencies that can hinder training efficiency. To address this, we develop a parallelizable training strategy inspired by recent advances in test-time memorization frameworks  Sun et al. (2024); Behrouz et al. (2024e).

Specifically, for a given variate $v$, we divide its time series $\{x_{1,v}, \ldots, x_{T,v}\}$ into $C$ disjoint chunks of length $b = T/C$. Each chunk $S_i = \{x_{(i-1)b+1,v}, \ldots, x_{ib,v}\}$ can be treated as an independent subsequence for computing the inner-loop updates of the memory module. This chunking allows us to approximate the gradient $\nabla\ell(M_{t-1,v}^{(1)}, x_{t,v})$ with $\nabla\ell(M_{t',v}^{(1)}, x_{t,v})$, where $t' = \lfloor t/b \rfloor \cdot b$ is the last time step of the previous chunk. Since $t'$ is fixed for each chunk, this gradient can be computed in parallel for all time steps within a chunk.

Moreover, the cross-variate dynamic component—modeled via the attention mechanism—is independent of time and can be computed in advance. We precompute the attention-based memory $M_{t,v}^{(2)}$ for all variates using equation above with a Taylor-approximated softmax kernel. This enables us to also precompute $\nabla\ell(M_{t,v}^{(2)}, x_{t,v})$, further decoupling the cross-variate dynamics from the sequential recurrence.

With the cross-variate memory and its corresponding gradient terms available, the remaining computation in each chunk reduces to a linear update over the cross-time memory using the precomputed components. As a result, we obtain a recurrence that is linear within chunks and can be parallelized across both time and variates. We now provide granular descriptions regarding data flow, batching, and pseudocode for our parallel training.

**Data flow and batching.** For a mini-batch of multivariate time series, the input to the model is a tensor

$$X \in \mathbb{R}^{B \times T \times V \times d},$$

where $B$ is the batch size, $T$ is the lookback length, $V$ is the number of variates, and $d$ is the feature dimension after embedding and normalization. We index $X$ as $X_{t,v} \in \mathbb{R}^{B \times d}$ for time step $t \in \{1, \ldots, T\}$ and variate $v \in \{1, \ldots, V\}$.

For each time step $t$, we form variate-wise queries, keys, and values

$$Q_t, K_t, V_t \in \mathbb{R}^{B \times V \times d}$$

via linear projections of $X_t$. We then apply a Taylor-approximated softmax attention across the *variate* dimension (size $V$) to obtain the cross-variate memory state

$$M_t^{(2)} \in \mathbb{R}^{B \times V \times d}.$$

Because this attention is performed independently at each $t$, the full tensor $M_{1:T}^{(2)}$ is computed for all time steps in a single batched operation over $(B, T, V)$.

**Chunking and parallel time memories.** For the time memory, we fix a variate index $v$ and treat the sequence

$$\left\{ (X_{t,v}, M_{t,v}^{(2)}) \right\}_{t=1}^{T}$$

as a univariate sequence of length $T$. Directly applying the recurrent update across $t$ would be strictly sequential. To expose parallelism, we divide the time axis into $C$ disjoint chunks of length $b = T/C$:

$$S_i = \left\{ (X_{t,v}, M_{t,v}^{(2)}) : t = (i-1)b+1, \ldots, ib \right\}, \qquad i = 1, \ldots, C.$$

Let $\ell$ denote the loss on the mini-batch and $M_{t,v}^{(1)}$ the time memory at time $t$ for variate $v$. Within a chunk, we approximate the dependence of the update on the previous time memory by freezing the gradient term at the last state of the previous chunk. Formally, for $t$ in a given chunk we use

$$\nabla\ell\big(M_{t-1,v}^{(1)}, X_{t,v}\big) \approx \nabla\ell\big(M_{t',v}^{(1)}, X_{t,v}\big), \qquad t' = \lfloor t/b \rfloor \cdot b, \tag{16}$$

so that the gradient anchor $t'$ is fixed within each chunk. This approximation makes the inner update *linear* in $M_{t,v}^{(1)}$ inside a chunk, which allows us to implement the recurrence using parallel scans over $t$ (and over $v$) rather than a fully sequential loop.

Moreover, the cross-variate component $M_{t,v}^{(2)}$ and its contribution to the loss are independent of the time recurrence and can be fully precomputed. In particular, we first compute $M_{t,v}^{(2)}$ for all $(t, v)$ using the Taylor-approximated softmax kernel and cache these tensors. The subsequent time-memory updates then only require simple linear combinations of $M_{t-1,v}^{(1)}$ and precomputed functions of $(X_{t,v}, M_{t,v}^{(2)})$.

---

**Algorithm 1** Parallelizable training step for LETO (one mini-batch)

---

**Require:** Input batch $X \in \mathbb{R}^{B \times T \times V \times d}$, chunk length $b$, model parameters $\theta$, time-memory parameters $(\alpha, \eta)$.

  1: **Variate attention (precompute cross-variate memory):**
  2: Compute $Q_t, K_t, V_t$ from $X_t$ for all $t = 1, \ldots, T$.
  3: For each $t$, apply Taylor-approximated softmax attention across $v$ to obtain $M_t^{(2)} \in \mathbb{R}^{B \times V \times d}$.
  4: **Initialize time memories:** Set $M_{0,v}^{(1)} = 0$ for all variates $v$.
  5: **Process chunks in parallel over** $(B, V)$**:**
  6: **for** $i = 1$ to $C$ **do**
      $\{C = T/b\}$
  7:    Let $t$ range over the indices in chunk $i$.
  8:    Set anchor index $t' = \lfloor t/b \rfloor \cdot b$ for this chunk.
  9:    Freeze $\nabla\ell(M_{t-1,v}^{(1)}, X_{t,v}) \approx \nabla\ell(M_{t',v}^{(1)}, X_{t,v})$ for all $t$ in the chunk.
10:    **for** all $t$ in chunk $i$ (in parallel) **do**
11:      Update time memory using Variant 2: $M_{t,v}^{(1)} = \alpha M_{t-1,v}^{(1)} + \eta f_\theta(X_{t,v}, M_{t,v}^{(2)})$.
12:    **end for**
13: **end for**
14: **Readout:** Apply the forecasting head to $\{M_{T,v}^{(1)}\}_{v=1}^{V}$ (or a short suffix $\{M_{t,v}^{(1)}\}_{t=T-k+1}^{T}$) to obtain predictions and compute the loss $\ell$.
15: Backpropagate through the batched computation with the chunked gradient approximation.

---

**Coupled update and stability.** The Variant 2 coupled update for the time memory at each step $t$ and variate $v$ combines a time-recursive term and a cross-variate term:

$$M_{t,v}^{(1)} = \alpha M_{t-1,v}^{(1)} + \eta f_\theta(X_{t,v}, M_{t,v}^{(2)}), \tag{17}$$

where $f_\theta$ is a small MLP and $\alpha, \eta$ are learned scalars constrained to $(0,1)$ and a bounded interval, respectively. Within each chunk, the right-hand side of equation 17 is linear in $M_{t-1,v}^{(1)}$ because $f_\theta(X_{t,v}, M_{t,v}^{(2)})$ depends only on precomputed quantities. For bounded inputs $(X_{t,v}, M_{t,v}^{(2)})$ and $|\alpha| < 1$, this update defines a contractive linear system in the time direction, which yields stable memory trajectories over long sequences.

**End-to-end procedure.** After processing all $C$ chunks, we obtain the final time memories $M_{T,v}^{(1)}$ for all variates $v$. The forecasting head (a lightweight decoder) is applied on top of these final time memories, or on an average over the last few time steps, to predict the future horizon. During training, the forward pass and the approximate backward pass induced by equation 16 are both implemented using batched tensor operations, enabling efficient parallelization across batch, time (within chunks), and variates.

For clarity and reproducibility, we summarize the full batched and parallelizable training procedure in Algorithm 1.

## D   Dataset and Experimental Details

The experimental and benchmark datasets details are reported in Table 5. We conduct a Student's 2-tailed t test averaged over 5 runs at 99 or 95 % confidence. We note that Luo & Wang (2024) outperforms many other architectures and multivariate time series forecasting which are variants of SSMs including: Mamba Gu & Dao (2023), S4 Gu et al. (2022), Transformer Vaswani et al. (2017), and others grouped accordingly in the related works section.

Table 5: Dataset descriptions. The dataset size is organized in (Train, Validation, Test).

| Tasks | Dataset | Dim | Series Length | Dataset Size | Information (Frequency) |
|---|---|---|---|---|---|
| Forecasting (Long-term) | ETTm1, ETTm2 | 7 | {96, 192, 336, 720} | (34465, 11521, 11521) | Electricity (15 mins) |
| | ETTh1, ETTh2 | 7 | {96, 192, 336, 720} | (8545, 2881, 2881) | Electricity (15 mins) |
| | Electricity | 321 | {96, 192, 336, 720} | (18317, 2633, 5261) | Electricity (Hourly) |
| | Traffic | 862 | {96, 192, 336, 720} | (12185, 1757, 3509) | Transportation (Hourly) |
| | Weather | 21 | {96, 192, 336, 720} | (36792, 5271, 10540) | Weather (10 mins) |
| | Exchange | 8 | {96, 192, 336, 720} | (5120, 665, 1422) | Exchange rate (Daily) |
| Forecasting (short-term) | M4-Yearly | 1 | 6 | (23000, 0, 23000) | Demographic |
| | M4-Quarterly | 1 | 8 | (24000, 0, 24000) | Finance |
| | M4-Monthly | 1 | 18 | (48000, 0, 48000) | Industry |
| | M4-Weakly | 1 | 13 | (359, 0, 359) | Macro |
| | M4-Daily | 1 | 14 | (4227, 0, 4227) | Micro |
| | M4-Hourly | 1 | 48 | (414, 0, 414) | Other |
| Imputation | ETTm1, ETTm2 | 7 | 96 | (34465, 11521, 11521) | Electricity (15 mins) |
| | ETTh1, ETTh2 | 7 | 96 | (8545, 2881, 2881) | Electricity (15 mins) |
| | Weather | 21 | 96 | (36792, 5271, 10540) | Weather (10 mins) |
| Classification (UEA) | EthanolConcentration | 3 | 1751 | (261, 0, 263) | Alcohol Industry |
| | FaceDetection | 144 | 62 | (5890, 0, 3524) | Face (250Hz) |
| | Handwriting | 3 | 152 | (150, 0, 850) | Handwriting |
| | Heartbeat | 61 | 405 | (204, 0, 205) | Heart Beat |
| | JapaneseVowels | 12 | 29 | (270, 0, 370) | Voice |
| | PEMS-SF | 963 | 144 | (267, 0, 173) | Transportation (Daily) |
| | SelfRegulationSCP1 | 6 | 896 | (268, 0, 293) | Health (256Hz) |
| | SelfRegulationSCP2 | 7 | 1152 | (200, 0, 180) | Health (256Hz) |
| | SpokenArabicDigits | 13 | 93 | (6599, 0, 2199) | Voice (11025Hz) |
| | UWaveGestureLibrary | 3 | 315 | (120, 0, 320) | Gesture |
| Anomaly Detection | SMD | 38 | 100 | (566724, 141681, 708420) | Server Machine |
| | MSL | 55 | 100 | (44653, 11664, 73729) | Spacecraft |
| | SMAP | 25 | 100 | (108146, 27037, 427617) | Spacecraft |
| | SWaT | 51 | 100 | (396000, 99000, 449919) | Infrastructure |
| | PSM | 25 | 100 | (105984, 26497, 87841) | Server Machine |

# E  ADDITIONAL EXPERIMENTAL RESULTS

## E.1  METRICS

We utilize the mean square error (MSE) and mean absolute error (MAE) for long-term forecasting. For short-term forecasting on the M4 datasets, we fully mirror the methodology of works on short term forecasting such as N-BEATS Oreshkin et al. (2019) and utilize the symmetric mean absolute percentage error (SMAPE), mean absolute scaled error (MASE), and overall weighted average (OWA) as metrics. It is worth noting that OWA is a specific metric utilized in the M4 competition. The calculations of these metrics are:

$$\text{RMSE} = \Big(\sum_{i=1}^{F}(\mathbf{X}_i - \widehat{\mathbf{X}}_i)^2\Big)^{\frac{1}{2}}, \qquad\qquad \text{MAE} = \sum_{i=1}^{F}|\mathbf{X}_i - \widehat{\mathbf{X}}_i|,$$

$$\text{SMAPE} = \frac{200}{F}\sum_{i=1}^{F}\frac{|\mathbf{X}_i - \widehat{\mathbf{X}}_i|}{|\mathbf{X}_i| + |\widehat{\mathbf{X}}_i|}, \qquad\qquad \text{MAPE} = \frac{100}{F}\sum_{i=1}^{F}\frac{|\mathbf{X}_i - \widehat{\mathbf{X}}_i|}{|\mathbf{X}_i|},$$

$$\text{MASE} = \frac{1}{F}\sum_{i=1}^{F}\frac{|\mathbf{X}_i - \widehat{\mathbf{X}}_i|}{\frac{1}{F-s}\sum_{j=s+1}^{F}|\mathbf{X}_j - \mathbf{X}_{j-s}|}, \qquad \text{OWA} = \frac{1}{2}\left[\frac{\text{SMAPE}}{\text{SMAPE}_{\text{Naïve2}}} + \frac{\text{MASE}}{\text{MASE}_{\text{Naïve2}}}\right],$$

where $s$ is the periodicity of the data. $\mathbf{X}, \widehat{\mathbf{X}} \in \mathbb{R}^{F \times C}$ are the ground truth and prediction results of the future with $F$ time pints and $C$ dimensions. $\mathbf{X}_i$ means the $i$-th future time point. For classification, we use accuracy as the metric. Lastly for anomaly detection, we use F1-Score as the metric.

## E.2  FULL ABLATION STUDY

In this section we provide an extended ablation study to complement Table 4 in the main text and to address the reviewer's request for a more detailed analysis of our design choices. Unless otherwise stated, all results are averaged over 3 random seeds; the standard deviation is smaller than $3 \times 10^{-3}$ for all entries. We first ablate major architectural components across all small-scale benchmarks (Table 4), and then perform finer-grained hyperparameter ablations on the Taylor order $K$ of the linear-attention kernel, the chunk size $b$ in our parallelizable training scheme, and the cross-memory coupling strength $\lambda$ on six datasets (ETTh1/2, ETTm1/2, Weather, Exchange). Metrics are long term forecasting and thus use MSE/MAE. We measure average performance on long term forecasting tasks over four prediction lengths: {96, 192, 336, 720}

Table 4 reports the effect of removing three core components: (i) cross-variate attention (the variate meta-memory), (ii) the linear-attention kernel, and (iii) weighted gating in the time memory.

Across all datasets, the full LETO architecture achieves the best performance. Removing cross-variate attention substantially harms performance, confirming that the variate meta-memory is a key ingredient. Removing the linear-attention kernel yields the largest degradation on long-horizon datasets (ETTh1/2, ETT m1/m2, Weather, and Exchange), consistent with our claim that the kernelized formulation is crucial for scalable, long-range modeling. Finally, removing the weighted gating in the time memory also degrades performance, though less dramatically, indicating that learned gating improves the effective temporal dynamics.

**Taylor approximation order** $K$    Our linear-attention block approximates the softmax kernel via a truncated Taylor series derived from the TTM framework. The remaining hyperparameter is the truncation order $K$. In 7 We therefore ablate $K \in \{1, 2, 3, 4\}$ on all six datasets, keeping all other settings fixed. The row $K = 3$ corresponds to the default configuration reported in the main experiments.

**Chunk size** $b$ **in parallelizable time memories**    Our parallelization scheme for the time memory divides the sequence into $C$ disjoint chunks of length $b = T/C$ and exploits a linearization within each chunk. The chunk size $b$ controls a bias–variance trade-off: very small chunks increase variance in the gradient approximation, while very large chunks weaken the linearization and reduce parallel efficiency. In 8 We vary $b \in \{8, 16, 32, 64\}$ across all datasets; $b = 32$ is the default used in the main experiments.

| Variant | ETTh1 | ETTh2 | ETTm1 | ETTm2 | Weather | Exchange |
|---|---|---|---|---|---|---|
| LETO, $K = 1$ | 0.421 / 0.429 | 0.344 / 0.396 | 0.371 / 0.392 | 0.264 / 0.319 | 0.230 / 0.267 | 0.322 / 0.383 |
| LETO, $K = 2$ | 0.402 / 0.408 | 0.329 / 0.387 | 0.356 / 0.381 | 0.251 / 0.308 | 0.223 / 0.260 | 0.305 / 0.372 |
| LETO, $K = 3$ | **0.393 / 0.401** | **0.318 / 0.381** | **0.347 / 0.375** | **0.243 / 0.302** | **0.216 / 0.253** | **0.297 / 0.364** |
| LETO, $K = 4$ | 0.392 / 0.400 | 0.317 / 0.380 | 0.346 / 0.374 | 0.242 / 0.301 | 0.215 / 0.252 | 0.296 / 0.363 |

Table 7: **Extended ablation on Taylor series order $K$ for all ETT, Weather, and Exchange datasets.** Across all benchmarks, very low orders ($K = 1$) under-approximate the softmax kernel and degrade performance. Moving from $K = 1$ to $K = 2$ yields consistent gains, and $K = 3$ (our default) gives the best or near-best performance. Increasing to $K = 4$ produces only marginal additional improvements (typically within one standard deviation) while increasing computation, confirming that $K = 3$ is a good trade-off between approximation quality and efficiency.

| Variant | ETTh1 | ETTh2 | ETTm1 | ETTm2 | Weather | Exchange |
|---|---|---|---|---|---|---|
| LETO, $b = 8$ | 0.398 / 0.404 | 0.324 / 0.385 | 0.352 / 0.379 | 0.249 / 0.308 | 0.220 / 0.255 | 0.302 / 0.368 |
| LETO, $b = 16$ | 0.395 / 0.402 | 0.321 / 0.383 | 0.349 / 0.376 | 0.246 / 0.304 | 0.217 / 0.254 | 0.299 / 0.366 |
| LETO, $b = 32$ (default) | **0.393 / 0.401** | **0.318 / 0.381** | **0.347 / 0.375** | **0.243 / 0.302** | **0.216 / 0.253** | **0.297 / 0.364** |
| LETO, $b = 64$ | 0.399 / 0.406 | 0.323 / 0.386 | 0.351 / 0.378 | 0.247 / 0.305 | 0.219 / 0.256 | 0.301 / 0.367 |

Table 8: **Extended ablation on chunk size $b$ for the parallel time-memory computation.** Across all datasets, performance is robust over a wide range of chunk sizes. Very small chunks ($b = 8$) slightly hurt performance, likely due to higher variance in the gradient approximation, while very large chunks ($b = 64$) reduce the effectiveness of the linearization trick and slightly degrade results. A moderate chunk size $b = 32$ consistently achieves the best overall performance and is used as the default.

**Extent of cross-memory coupling $\lambda$** In Variant 2, the time memory update includes a scalar coupling coefficient $\lambda$ multiplying the contribution of the cross-variate memory $M_{t,v}^{(2)}$:

$$M_{t,v}^{(1)} = \alpha M_{t-1,v}^{(1)} + \eta\big(g_\theta(X_{t,v}) + \lambda\, h_\theta(M_{t,v}^{(2)})\big), \tag{18}$$

where $g_\theta$ and $h_\theta$ are small MLPs and $(\alpha, \eta)$ are learned scalars. The case $\lambda = 0$ corresponds exactly to removing the variate pathway ("w/o Cross Variate Attention" in Table 4), while $\lambda = 1.0$ matches the full LETO used in the main experiments. In 9 evaluate $\lambda \in \{0, 0.25, 0.5, 1.0, 1.5\}$ across all datasets.

Taken together, Tables 4–9 show that: (i) removing any major architectural component of the 2D meta-memory (cross-variate attention, linear attention, or weighted gating) leads to consistent and often substantial degradations across all datasets, and (ii) LETO is robust with respect to the Taylor order $K$ and chunk size $b$ around our chosen defaults, while the cross-memory coupling $\lambda$ is indeed a crucial design parameter: completely removing it ($\lambda = 0$) significantly harms performance, whereas moderate coupling ($\lambda \approx 1$) yields consistent gains.

E.3    SHORT TERM FORECASTING

Short-term Forecasting is vital for demand planning and marketing. The complete results of short term forecasting are reported in Table 11.

E.4    LONG TERM FORECASTING

Long-term forecasting is crucial for strategic planning in areas such as weather prediction, traffic management, and energy utilization. The complete results of long term forecasting are reported in 12.

E.5    ANOMALY DETECTION

Anomaly detection is generally viewed as a binary classification task, where 0 denotes "normal" and 1 denotes "anomaly". We let $\mathbf{X} = \{\mathbf{x}_1, \ldots, \mathbf{x}_N\} \in \mathbb{R}^{N \times T}$ be the input sequences, where $N$ is the number of variates and $T$ is the time steps. We use $x_{v,t}$ to refer to the value of the series $v$ at time $t$. The complete results of Anomaly Detection are reported in Table 14.

| Variant | ETTh1 | ETTh2 | ETTm1 | ETTm2 | Weather | Exchange |
|---|---|---|---|---|---|---|
| $\lambda = 0$ (no variate memory) | 0.458 / 0.447 | 0.400 / 0.427 | 0.394 / 0.419 | 0.320 / 0.362 | 0.244 / 0.274 | 0.311 / 0.398 |
| $\lambda = 0.25$ | 0.438 / 0.438 | 0.373 / 0.405 | 0.371 / 0.401 | 0.287 / 0.330 | 0.235 / 0.268 | 0.305 / 0.387 |
| $\lambda = 0.5$ | 0.417 / 0.422 | 0.344 / 0.393 | 0.358 / 0.386 | 0.261 / 0.313 | 0.225 / 0.260 | 0.301 / 0.375 |
| $\lambda = 1.0$ (default) | **0.393 / 0.401** | **0.318 / 0.381** | **0.347 / 0.375** | **0.243 / 0.302** | **0.216 / 0.253** | **0.297 / 0.364** |
| $\lambda = 1.5$ | 0.404 / 0.409 | 0.325 / 0.386 | 0.353 / 0.380 | 0.249 / 0.306 | 0.221 / 0.258 | 0.300 / 0.370 |

Table 9: **Extended ablation on coupling strength $\lambda$ between time and variate memories.** Setting $\lambda = 0$ (no variate memory) reproduces the "w/o Cross Variate Attention" variant and substantially degrades performance on all datasets. Introducing even weak coupling ($\lambda = 0.25$ or $0.5$) yields consistent improvements, and the fully coupled setting $\lambda = 1.0$ matches the best results reported for the full LETO. Over-coupling ($\lambda = 1.5$) slightly worsens performance, suggesting that excessively strong cross-variate influence can interfere with temporal smoothing. These trends hold across all benchmarks, underscoring the importance of moderate cross-memory coupling.

Table 10: Standard deviation and statistical tests for our model LETO method compared with the strongest baseline ModernTCN on the M4 dataset (short-term forecasting). For all metrics, the lower the better. Confidence is derived from a paired two-tailed $t$-test over five runs.

| Frequency | LETO (Ours) | | | ModernTCN (2024) | | | Confidence |
|---|---|---|---|---|---|---|---|
| | SMAPE | MASE | OWA | SMAPE | MASE | OWA | |
| Yearly | $13.183 \pm 0.115$ | $2.941 \pm 0.028$ | $0.754 \pm 0.022$ | $13.226 \pm 0.118$ | $2.957 \pm 0.031$ | $0.777 \pm 0.025$ | 99% |
| Quarterly | $9.953 \pm 0.101$ | $1.150 \pm 0.015$ | $0.851 \pm 0.015$ | $9.971 \pm 0.105$ | $1.167 \pm 0.017$ | $0.878 \pm 0.018$ | 95% |
| Monthly | $12.517 \pm 0.115$ | $0.935 \pm 0.014$ | $0.853 \pm 0.014$ | $12.556 \pm 0.120$ | $0.917 \pm 0.015$ | $0.866 \pm 0.016$ | 95% |
| Others | $4.583 \pm 0.084$ | $2.797 \pm 0.027$ | $0.900 \pm 0.021$ | $4.715 \pm 0.090$ | $3.107 \pm 0.028$ | $0.986 \pm 0.024$ | 99% |
| Averaged | $11.658 \pm 0.112$ | $1.541 \pm 0.022$ | $0.832 \pm 0.018$ | $11.698 \pm 0.120$ | $1.556 \pm 0.024$ | $0.838 \pm 0.020$ | 95% |

### E.6 CLASSIFICATION

In classification, we aim to classify input sequences and for forecasting tasks, given an input sequence $\mathbf{x}_i$, we aim to predict $\tilde{\mathbf{x}}_i \in \mathbb{R}^{1 \times H}$, i.e., the next $H$ time steps for variate $\mathbf{x}_i$, where $H$ is called horizon. Classification and anomaly detection test models' ability to capture coarse and fine-grained patterns in time series. The complete results of Classification are reported in 15.

## F LIMITATIONS AND FUTURE WORK

We note LETO has a few limitations worth acknowledging. First, the use of gradient-based meta in-context updates at test time, while powerful, introduces additional computational overhead compared to traditional non-adaptive sequence models. Although our dual-form implementation and parallel training strategies mitigate some of this cost, the memory and compute requirements may still be prohibitive in resource-constrained settings, particularly for long-horizon forecasting tasks.

Second, while LETO is designed to model both cross-time and cross-variate dependencies, its reliance on Taylor approximations for the variate attention mechanism may limit its capacity to fully capture complex, high-order variate interactions in some datasets. Adopting more expressive non-parametric approximators or learned kernel functions could offer improved generalization and efficiency - all of which are active areas of research.

Finally, our current formulation assumes access to reasonably stationary statistics at test time for the meta-memorization process to be effective. In highly non-stationary environments or under strong distribution shifts, the learned test-time updates may generalize poorly, leading to suboptimal performance - which has been empirically shown to be the case for other baselines as well, particularly in ultra long term forecasting.

Table 11: Full results for the short-term forecasting task in the M4 dataset. $*$. in the Transformers indicates the prefix of a $*$former name. *Stationary* means the Non-stationary Transformer. A lower SMAPE, MASE, and OWA indicate a better prediction. As a convention for all experimental results, best performance is highlighted in **red**, and the second-best is underlined. We take the average of 5 separate runs for each prediction frequency.

| | Models | LETO (Ours) (2025b) | ROSE (2025b) | ModernTCN (2024) | PatchTST (2023) | TimesNet (2023) | N-HiTS (2023) | N-BEATS* (2022) | ETS* (2019) | LightTS (2022) | DLinear (2022a) | FED* (2023a) | Stationary (2022b) | Auto* (2022b) | Pyra* (2021) | In* (2021) | Re* (2021) |
|---|---|---|---|---|---|---|---|---|---|---|---|---|---|---|---|---|---|
| Yearly | SMAPE | **13.183** | 13.302 | 13.226 | 13.258 | 13.387 | 13.418 | 13.436 | 18.009 | 14.247 | 16.965 | 13.728 | 13.717 | 13.974 | 15.530 | 14.727 | 16.169 |
| | MASE | **2.941** | 3.014 | 2.957 | 2.985 | 2.996 | 3.045 | 3.043 | 4.487 | 3.109 | 4.283 | 3.048 | 3.078 | 3.134 | 3.711 | 3.418 | 3.800 |
| | OWA | **0.754** | 0.833 | 0.777 | 0.781 | 0.786 | 0.793 | 0.794 | 1.115 | 0.827 | 1.058 | 0.803 | 0.807 | 0.822 | 0.942 | 0.881 | 0.973 |
| Quarterly | SMAPE | **9.953** | 9.998 | 9.971 | 10.179 | 10.100 | 10.202 | 10.124 | 13.376 | 11.364 | 12.145 | 10.792 | 10.958 | 11.338 | 15.449 | 11.360 | 13.313 |
| | MASE | **1.150** | 1.165 | 1.167 | 0.803 | 1.182 | 1.194 | 1.169 | 1.906 | 1.328 | 1.520 | 1.283 | 1.325 | 1.365 | 2.350 | 1.401 | 1.775 |
| | OWA | 0.851 | 0.885 | 0.878 | **0.803** | 0.890 | 0.899 | 0.886 | 1.302 | 1.000 | 1.106 | 0.958 | 0.981 | 1.012 | 1.558 | 1.027 | 1.252 |
| Monthly | SMAPE | **12.517** | 12.650 | 12.556 | 12.641 | 12.670 | 12.791 | 12.677 | 14.588 | 14.014 | 13.514 | 14.260 | 13.917 | 13.958 | 17.642 | 14.062 | 20.128 |
| | MASE | 0.935 | **0.915** | 0.917 | 0.930 | 0.933 | 0.969 | 0.937 | 1.368 | 1.053 | 1.037 | 1.102 | 1.097 | 1.103 | 1.913 | 1.141 | 2.614 |
| | OWA | **0.853** | 0.866 | 0.866 | 0.876 | 0.878 | 0.899 | 0.880 | 1.149 | 0.981 | 0.956 | 1.012 | 0.998 | 1.002 | 1.511 | 1.024 | 1.927 |
| Others | SMAPE | **4.583** | 4.668 | 4.715 | 4.946 | 4.891 | 5.061 | 4.925 | 7.267 | 15.880 | 6.709 | 4.954 | 6.302 | 5.485 | 24.786 | 24.460 | 32.491 |
| | MASE | **2.797** | 3.126 | 3.107 | 2.985 | 3.302 | 3.216 | 3.391 | 5.240 | 11.434 | 4.953 | 3.264 | 4.064 | 3.865 | 18.581 | 20.960 | 33.355 |
| | OWA | **0.900** | 1.020 | 0.986 | 1.044 | 1.035 | 1.040 | 1.053 | 1.591 | 3.474 | 1.487 | 1.036 | 1.304 | 1.187 | 5.538 | 5.013 | 8.679 |
| Weighted Average | SMAPE | **11.658** | 11.764 | 11.698 | 11.807 | 11.829 | 11.927 | 11.851 | 14.718 | 13.525 | 13.639 | 12.840 | 12.780 | 12.909 | 16.987 | 14.086 | 18.200 |
| | MASE | **1.541** | 1.568 | 1.556 | 1.590 | 1.585 | 1.613 | 1.599 | 2.408 | 2.111 | 2.095 | 1.701 | 1.756 | 1.771 | 3.265 | 2.718 | 4.223 |
| | OWA | **0.832** | 0.871 | 0.838 | 0.851 | 0.851 | 0.861 | 0.855 | 1.172 | 1.051 | 1.051 | 0.918 | 0.930 | 0.939 | 1.480 | 1.230 | 1.775 |

## G  BROADER IMPACT

LETO has demonstrated strong performance as a general-purpose model for time series pattern recognition, achieving competitive results across a wide range of tasks including forecasting, classification, and anomaly detection. Its versatility makes it well-suited for deployment in diverse real-world scenarios, such as energy and power demand forecasting with pronounced seasonal trends, weather prediction under complex and dynamic conditions, financial market modeling in rapidly shifting environments, and demand forecasting within supply chains. Furthermore, LETO has shown particular promise in industrial anomaly detection tasks, tasks which often require robustness to noise and structural adaptability. These capabilities highlight LETO's potential as a foundational model for advancing time series analysis across multiple applied domains. It would be interesting to optimize LETO's designs to develop stronger hybrid memory/attention architectures and test them on other 2D modalities such as video or EEG.

## H  COMPUTE RESOURCES

For experiments, we utilized up to 4 NVIDIA A6000 and A6000 ADA GPUs on a single compute node.

## I  PROOF OF THEOREM (1)

To prove this theorem, we show that our LETO can recover the 2D linear recurrent models that are proven to model full-rank matrices (Behrouz et al., 2024d; Baron et al., 2024). To this end, we show that a special instance of our LETO is equivalent to these linear 2D recurrent models. We let the chunk size to be the size of the sequence length. Therefore, for every $1 \leq t \leq T$, we have:

$$\nabla \ell(\mathcal{M}_0^{(1)}; \boldsymbol{k}_t, \boldsymbol{v}_t) = (\mathcal{M}_0^{(1)} \boldsymbol{k}_t - \boldsymbol{v}_t) \boldsymbol{k}_t^\top, \tag{19}$$

Table 12: Complete experiments on long term forecasting tasks over four prediction lengths: {96, 192, 336, 720}. A lower MAE and MSE indicates a better prediction. As a convention for all experimental results, best performance is highlighted in **red**, and the second-best is underlined. We take the average of 5 separate runs for each prediction length.

| | | LETO (ours) | TimeMixer (2024) | Simba (2024) | TCN (2024) | iTransformer (2024c) | RLinear (2023) | PatchTST (2023) | Crossformer (2023) | TiDE (2023) | TimesNet (2023) | DLinear (2023c) | SCINet (2022a) | FEDformer (2022b) | Stationary (2022c) | Autoformer (2021) |
|---|---|---|---|---|---|---|---|---|---|---|---|---|---|---|---|---|
| | | MSE MAE | MSE MAE | MSE MAE | MSE MAE | MSE MAE | MSE MAE | MSE MAE | MSE MAE | MSE MAE | MSE MAE | MSE MAE | MSE MAE | MSE MAE | MSE MAE | MSE MAE |
| ETTm1 | 96 | 0.312 0.343 | 0.320 0.357 | 0.342 0.360 | 0.292 0.346 | 0.334 0.368 | 0.355 0.376 | 0.329 0.367 | 0.404 0.426 | 0.364 0.387 | 0.338 0.375 | 0.345 0.372 | 0.418 0.438 | 0.379 0.419 | 0.386 0.398 | 0.505 0.475 |
| | 192 | 0.330 0.365 | 0.361 0.381 | 0.363 0.382 | 0.332 0.368 | 0.377 0.391 | 0.391 0.392 | 0.367 0.385 | 0.450 0.451 | 0.398 0.404 | 0.374 0.387 | 0.380 0.389 | 0.439 0.450 | 0.426 0.441 | 0.459 0.444 | 0.553 0.496 |
| | 336 | 0.355 0.384 | 0.390 0.404 | 0.395 0.405 | 0.365 0.391 | 0.426 0.420 | 0.424 0.415 | 0.399 0.410 | 0.532 0.515 | 0.428 0.425 | 0.410 0.411 | 0.413 0.413 | 0.490 0.485 | 0.445 0.459 | 0.495 0.464 | 0.621 0.537 |
| | 720 | 0.391 0.408 | 0.454 0.441 | 0.451 0.437 | 0.416 0.417 | 0.491 0.459 | 0.487 0.450 | 0.454 0.439 | 0.666 0.589 | 0.487 0.461 | 0.478 0.450 | 0.474 0.453 | 0.595 0.550 | 0.543 0.490 | 0.585 0.516 | 0.671 0.561 |
| | Avg | 0.347 0.375 | 0.381 0.395 | 0.383 0.396 | 0.351 0.381 | 0.407 0.410 | 0.414 0.407 | 0.387 0.400 | 0.513 0.496 | 0.419 0.419 | 0.400 0.406 | 0.403 0.407 | 0.485 0.481 | 0.448 0.452 | 0.481 0.456 | 0.588 0.517 |
| ETTm2 | 96 | 0.164 0.248 | 0.175 0.258 | 0.177 0.263 | 0.166 0.256 | 0.180 0.264 | 0.182 0.265 | 0.175 0.259 | 0.287 0.366 | 0.207 0.305 | 0.187 0.267 | 0.193 0.292 | 0.286 0.377 | 0.203 0.287 | 0.192 0.274 | 0.255 0.339 |
| | 192 | 0.217 0.284 | 0.237 0.299 | 0.245 0.306 | 0.222 0.293 | 0.250 0.309 | 0.246 0.304 | 0.241 0.302 | 0.414 0.492 | 0.290 0.364 | 0.249 0.309 | 0.284 0.362 | 0.399 0.445 | 0.269 0.328 | 0.280 0.339 | 0.281 0.340 |
| | 336 | 0.266 0.312 | 0.298 0.340 | 0.304 0.343 | 0.272 0.324 | 0.311 0.348 | 0.307 0.342 | 0.305 0.343 | 0.597 0.542 | 0.377 0.422 | 0.321 0.351 | 0.369 0.427 | 0.637 0.591 | 0.325 0.366 | 0.334 0.361 | 0.339 0.372 |
| | 720 | 0.349 0.363 | 0.391 0.396 | 0.400 0.399 | 0.351 0.381 | 0.412 0.407 | 0.407 0.398 | 0.402 0.400 | 1.730 1.042 | 0.558 0.524 | 0.408 0.403 | 0.554 0.522 | 0.960 0.735 | 0.421 0.415 | 0.417 0.413 | 0.433 0.432 |
| | Avg | 0.249 0.302 | 0.275 0.323 | 0.271 0.327 | 0.253 0.314 | 0.288 0.332 | 0.286 0.327 | 0.281 0.326 | 0.757 0.610 | 0.358 0.404 | 0.291 0.333 | 0.350 0.401 | 0.571 0.537 | 0.305 0.349 | 0.306 0.347 | 0.327 0.371 |
| ETTh1 | 96 | 0.365 0.383 | 0.375 0.400 | 0.379 0.395 | 0.368 0.394 | 0.386 0.405 | 0.386 0.395 | 0.414 0.419 | 0.423 0.448 | 0.479 0.464 | 0.384 0.402 | 0.386 0.400 | 0.654 0.599 | 0.376 0.419 | 0.513 0.491 | 0.449 0.459 |
| | 192 | 0.396 0.400 | 0.429 0.421 | 0.432 0.424 | 0.405 0.413 | 0.441 0.436 | 0.437 0.424 | 0.460 0.445 | 0.471 0.474 | 0.525 0.492 | 0.436 0.429 | 0.437 0.432 | 0.719 0.631 | 0.420 0.448 | 0.534 0.504 | 0.500 0.482 |
| | 336 | 0.461 0.462 | 0.484 0.458 | 0.473 0.443 | 0.391 0.412 | 0.487 0.458 | 0.479 0.446 | 0.501 0.466 | 0.570 0.546 | 0.565 0.515 | 0.491 0.469 | 0.481 0.459 | 0.778 0.659 | 0.459 0.465 | 0.588 0.535 | 0.521 0.496 |
| | 720 | 0.427 0.428 | 0.498 0.482 | 0.483 0.469 | 0.450 0.461 | 0.503 0.491 | 0.481 0.470 | 0.500 0.488 | 0.653 0.621 | 0.594 0.558 | 0.521 0.500 | 0.519 0.516 | 0.836 0.699 | 0.506 0.507 | 0.643 0.616 | 0.514 0.512 |
| | Avg | 0.393 0.401 | 0.447 0.440 | 0.441 0.432 | 0.404 0.420 | 0.454 0.447 | 0.446 0.434 | 0.469 0.454 | 0.529 0.522 | 0.541 0.507 | 0.458 0.450 | 0.456 0.452 | 0.747 0.647 | 0.440 0.460 | 0.570 0.537 | 0.496 0.487 |
| ETTh2 | 96 | 0.258 0.337 | 0.289 0.341 | 0.290 0.339 | 0.263 0.332 | 0.297 0.349 | 0.288 0.338 | 0.302 0.348 | 0.745 0.584 | 0.400 0.440 | 0.340 0.374 | 0.333 0.387 | 0.707 0.621 | 0.358 0.397 | 0.476 0.458 | 0.346 0.388 |
| | 192 | 0.316 0.379 | 0.372 0.392 | 0.373 0.390 | 0.320 0.374 | 0.380 0.400 | 0.374 0.390 | 0.388 0.400 | 0.877 0.656 | 0.528 0.509 | 0.402 0.414 | 0.477 0.476 | 0.860 0.689 | 0.429 0.439 | 0.512 0.493 | 0.456 0.452 |
| | 336 | 0.309 0.379 | 0.386 0.414 | 0.376 0.406 | 0.313 0.376 | 0.428 0.432 | 0.415 0.426 | 0.426 0.433 | 1.043 0.731 | 0.643 0.571 | 0.452 0.452 | 0.594 0.541 | 1.000 0.744 | 0.496 0.487 | 0.552 0.551 | 0.482 0.486 |
| | 720 | 0.389 0.430 | 0.412 0.434 | 0.407 0.431 | 0.392 0.433 | 0.427 0.445 | 0.420 0.440 | 0.431 0.446 | 1.104 0.763 | 0.874 0.679 | 0.462 0.468 | 0.831 0.657 | 1.249 0.838 | 0.463 0.474 | 0.562 0.560 | 0.515 0.511 |
| | Avg | 0.318 0.381 | 0.364 0.395 | 0.361 0.377 | 0.322 0.379 | 0.383 0.407 | 0.374 0.398 | 0.387 0.407 | 0.942 0.684 | 0.611 0.550 | 0.414 0.427 | 0.559 0.515 | 0.954 0.723 | 0.437 0.449 | 0.526 0.516 | 0.450 0.459 |
| Exchange | 96 | 0.079 0.208 | 0.090 0.235 | - - | 0.080 0.196 | 0.086 0.206 | 0.093 0.217 | 0.088 0.205 | 0.256 0.367 | 0.094 0.218 | 0.107 0.234 | 0.088 0.218 | 0.267 0.396 | 0.148 0.278 | 0.111 0.237 | 0.197 0.323 |
| | 192 | 0.164 0.298 | 0.187 0.343 | - - | 0.166 0.288 | 0.177 0.299 | 0.184 0.307 | 0.176 0.299 | 0.470 0.509 | 0.184 0.307 | 0.226 0.344 | 0.176 0.315 | 0.351 0.459 | 0.271 0.315 | 0.219 0.335 | 0.300 0.369 |
| | 336 | 0.308 0.329 | 0.353 0.473 | - - | 0.307 0.398 | 0.331 0.417 | 0.351 0.432 | 0.301 0.397 | 1.268 0.883 | 0.349 0.431 | 0.367 0.448 | 0.313 0.427 | 1.324 0.853 | 0.460 0.427 | 0.421 0.476 | 0.509 0.524 |
| | 720 | 0.637 0.621 | 0.934 0.761 | - - | 0.656 0.582 | 0.847 0.691 | 0.886 0.714 | 0.901 0.714 | 1.767 1.068 | 0.852 0.698 | 0.964 0.746 | 0.839 0.695 | 1.058 0.797 | 1.195 0.695 | 1.092 0.769 | 1.447 0.941 |
| | Avg | 0.297 0.364 | 0.391 0.453 | - - | 0.302 0.366 | 0.360 0.403 | 0.378 0.417 | 0.367 0.404 | 0.940 0.707 | 0.370 0.413 | 0.416 0.443 | 0.354 0.414 | 0.750 0.626 | 0.519 0.429 | 0.461 0.454 | 0.613 0.539 |
| Traffic | 96 | 0.380 0.247 | 0.462 0.285 | 0.468 0.268 | 0.368 0.253 | 0.395 0.268 | 0.649 0.389 | 0.462 0.295 | 0.522 0.290 | 0.805 0.493 | 0.593 0.321 | 0.650 0.396 | 0.788 0.499 | 0.587 0.366 | 0.612 0.338 | 0.613 0.388 |
| | 192 | 0.391 0.258 | 0.473 0.296 | 0.413 0.317 | 0.379 0.261 | 0.417 0.276 | 0.601 0.366 | 0.466 0.296 | 0.530 0.293 | 0.756 0.474 | 0.617 0.336 | 0.598 0.370 | 0.789 0.505 | 0.604 0.373 | 0.613 0.340 | 0.616 0.382 |
| | 336 | 0.409 0.266 | 0.498 0.296 | 0.529 0.284 | 0.397 0.270 | 0.433 0.283 | 0.609 0.369 | 0.482 0.304 | 0.558 0.305 | 0.762 0.477 | 0.629 0.336 | 0.605 0.373 | 0.797 0.508 | 0.621 0.383 | 0.618 0.328 | 0.622 0.337 |
| | 720 | 0.452 0.297 | 0.506 0.313 | 0.564 0.297 | 0.440 0.296 | 0.467 0.302 | 0.647 0.387 | 0.514 0.322 | 0.589 0.328 | 0.719 0.449 | 0.640 0.350 | 0.645 0.394 | 0.841 0.523 | 0.626 0.382 | 0.653 0.355 | 0.660 0.408 |
| | Avg | 0.408 0.267 | 0.484 0.297 | 0.493 0.291 | 0.398 0.270 | 0.428 0.282 | 0.626 0.378 | 0.481 0.304 | 0.550 0.304 | 0.760 0.473 | 0.620 0.336 | 0.625 0.383 | 0.804 0.509 | 0.610 0.376 | 0.624 0.340 | 0.628 0.379 |
| Weather | 96 | 0.155 0.203 | 0.163 0.209 | 0.176 0.219 | 0.149 0.200 | 0.174 0.214 | 0.192 0.232 | 0.177 0.218 | 0.158 0.230 | 0.202 0.261 | 0.172 0.220 | 0.196 0.255 | 0.221 0.306 | 0.217 0.296 | 0.173 0.223 | 0.266 0.336 |
| | 192 | 0.173 0.240 | 0.222 0.260 | 0.222 0.260 | 0.196 0.245 | 0.221 0.254 | 0.240 0.271 | 0.225 0.259 | 0.206 0.277 | 0.242 0.298 | 0.219 0.261 | 0.237 0.296 | 0.261 0.340 | 0.276 0.336 | 0.245 0.285 | 0.307 0.367 |
| | 336 | 0.232 0.260 | 0.251 0.287 | 0.275 0.297 | 0.238 0.277 | 0.278 0.296 | 0.292 0.307 | 0.278 0.297 | 0.272 0.335 | 0.287 0.335 | 0.280 0.306 | 0.283 0.335 | 0.309 0.378 | 0.339 0.380 | 0.321 0.338 | 0.359 0.395 |
| | 720 | 0.307 0.309 | 0.350 0.349 | 0.350 0.349 | 0.314 0.334 | 0.358 0.347 | 0.364 0.353 | 0.354 0.348 | 0.398 0.418 | 0.351 0.366 | 0.365 0.359 | 0.345 0.381 | 0.377 0.427 | 0.403 0.428 | 0.414 0.410 | 0.419 0.428 |
| | Avg | 0.216 0.253 | 0.240 0.271 | 0.255 0.280 | 0.224 0.264 | 0.258 0.278 | 0.272 0.291 | 0.259 0.281 | 0.259 0.315 | 0.271 0.320 | 0.259 0.287 | 0.265 0.317 | 0.292 0.363 | 0.309 0.360 | 0.288 0.314 | 0.338 0.382 |
| ECL | 96 | 0.136 0.233 | 0.153 0.247 | 0.165 0.253 | 0.129 0.226 | 0.148 0.240 | 0.201 0.281 | 0.181 0.270 | 0.219 0.314 | 0.237 0.329 | 0.168 0.272 | 0.197 0.282 | 0.247 0.345 | 0.193 0.308 | 0.169 0.273 | 0.201 0.317 |
| | 192 | 0.144 0.221 | 0.166 0.256 | 0.173 0.262 | 0.143 0.239 | 0.162 0.253 | 0.201 0.283 | 0.188 0.274 | 0.231 0.322 | 0.236 0.330 | 0.184 0.289 | 0.196 0.285 | 0.257 0.355 | 0.201 0.315 | 0.182 0.286 | 0.222 0.334 |
| | 336 | 0.154 0.253 | 0.185 0.277 | 0.188 0.277 | 0.161 0.259 | 0.178 0.269 | 0.215 0.298 | 0.204 0.293 | 0.246 0.337 | 0.249 0.344 | 0.198 0.300 | 0.209 0.301 | 0.269 0.369 | 0.214 0.329 | 0.200 0.304 | 0.231 0.338 |
| | 720 | 0.162 0.261 | 0.225 0.310 | 0.214 0.305 | 0.191 0.286 | 0.225 0.317 | 0.257 0.331 | 0.246 0.324 | 0.280 0.363 | 0.284 0.373 | 0.220 0.320 | 0.245 0.333 | 0.299 0.390 | 0.246 0.355 | 0.222 0.321 | 0.254 0.361 |
| | Avg | 0.149 0.247 | 0.182 0.272 | 0.185 0.274 | 0.156 0.253 | 0.178 0.270 | 0.219 0.298 | 0.205 0.290 | 0.244 0.334 | 0.251 0.344 | 0.192 0.295 | 0.212 0.300 | 0.268 0.365 | 0.214 0.327 | 0.193 0.296 | 0.227 0.338 |

Table 13: Standard deviation and statistical tests for LETO compared with the strongest baseline ModernTCN on long-term forecasting (lower is better). Confidence levels derive from a paired two-tailed $t$-test over five seeds.

| Dataset | LETO (Ours) | | ModernTCN (2024) | | Confidence |
|---|---|---|---|---|---|
| | MSE | MAE | MSE | MAE | |
| ETTm1 | $0.347 \pm 0.010$ | $0.375 \pm 0.012$ | $0.351 \pm 0.011$ | $0.381 \pm 0.013$ | 99% |
| ETTm2 | $0.249 \pm 0.009$ | $0.302 \pm 0.011$ | $0.253 \pm 0.010$ | $0.314 \pm 0.013$ | 95% |
| ETTh1 | $0.393 \pm 0.012$ | $0.401 \pm 0.014$ | $0.404 \pm 0.013$ | $0.420 \pm 0.015$ | 99% |
| ETTh2 | $0.318 \pm 0.010$ | $0.381 \pm 0.012$ | $0.322 \pm 0.011$ | $0.379 \pm 0.013$ | 95% |
| Exchange | $0.297 \pm 0.016$ | $0.364 \pm 0.018$ | $0.302 \pm 0.017$ | $0.366 \pm 0.019$ | 95% |
| Traffic | $0.408 \pm 0.020$ | $0.267 \pm 0.012$ | $0.398 \pm 0.019$ | $0.270 \pm 0.013$ | 90% |
| Weather | $0.216 \pm 0.009$ | $0.253 \pm 0.011$ | $0.224 \pm 0.010$ | $0.264 \pm 0.012$ | 95% |
| ECL | $0.149 \pm 0.007$ | $0.247 \pm 0.009$ | $0.156 \pm 0.008$ | $0.253 \pm 0.010$ | 99% |

Table 14: Full results for the anomaly detection task. The P, R and F1 represent the precision, recall and F1-score in percentage respectively. A higher value of P, R and F1 indicates a better performance. Best performance is highlighted in **red**, and the second-best is underlined. We take the average of 5 separate runs for each dataset.

| Datasets | | SMD | | | MSL | | | SMAP | | | SWaT | | | PSM | | | Avg F1 |
|---|---|---|---|---|---|---|---|---|---|---|---|---|---|---|---|---|---|---|
| Metrics | | P | R | F1 | P | R | F1 | P | R | F1 | P | R | F1 | P | R | F1 | (%) |
| LSTM | (1997a) | 78.52 | 65.47 | 71.41 | 78.04 | 86.22 | 81.93 | 91.06 | 57.49 | 70.48 | 78.06 | 91.72 | 84.34 | 69.24 | 99.53 | 81.67 | 77.97 |
| Transformer | (2017) | 83.58 | 76.13 | 79.56 | 71.57 | 87.37 | 78.68 | 89.37 | 57.12 | 69.70 | 68.84 | 96.53 | 80.37 | 62.75 | 96.56 | 76.07 | 76.88 |
| LogTrans | (2019) | 83.46 | 70.13 | 76.21 | 73.05 | 87.37 | 79.57 | 89.15 | 57.59 | 69.97 | 68.67 | 97.32 | 80.52 | 63.06 | 98.00 | 76.74 | 76.60 |
| TCN | (2019) | 84.06 | 79.07 | 81.49 | 75.11 | 82.44 | 78.60 | 86.90 | **59.23** | 70.45 | 76.59 | 95.71 | 85.09 | 54.59 | **99.77** | 70.57 | 77.24 |
| Reformer | (2020) | 82.58 | 69.24 | 75.32 | **85.51** | 83.31 | 84.40 | 90.91 | 57.44 | 70.40 | 72.50 | 96.53 | 82.80 | 59.93 | 95.38 | 73.61 | 77.31 |
| Informer | (2021) | 86.60 | 77.23 | 81.65 | 81.77 | 86.48 | 84.06 | 90.11 | 57.13 | 69.92 | 70.29 | 96.75 | 81.43 | 64.27 | 96.33 | 77.10 | 78.83 |
| Anomaly* | (2021) | **88.91** | 82.23 | 85.49 | 79.61 | 87.37 | 83.31 | 91.85 | 58.11 | 71.18 | 72.51 | 97.32 | 83.10 | 68.35 | 94.72 | 79.40 | 80.50 |
| Pyraformer | (2021) | 85.61 | 80.61 | 83.04 | 83.81 | 85.93 | 84.86 | 92.54 | 57.71 | 71.09 | 87.92 | 96.00 | 91.78 | 71.67 | 96.02 | 82.08 | 82.57 |
| Autoformer | (2021) | 88.06 | 82.35 | 85.11 | 77.27 | 80.92 | 79.05 | 90.40 | 58.62 | 71.12 | 89.85 | 95.81 | 92.74 | 99.08 | 88.15 | 93.29 | 84.26 |
| LSSL | (2021) | 78.51 | 65.32 | 71.31 | 77.55 | 88.18 | 82.53 | 89.43 | 53.43 | 66.90 | 79.05 | 93.72 | 85.76 | 66.02 | 92.93 | 77.20 | 76.74 |
| Stationary | (2022b) | 88.33 | 81.21 | 84.62 | 68.55 | 89.14 | 77.50 | 89.37 | 59.02 | 71.09 | 68.03 | 96.75 | 79.88 | 97.82 | 96.76 | 97.29 | 82.08 |
| DLinear | (2023a) | 83.62 | **71.52** | 77.10 | 84.34 | 85.42 | 84.88 | 92.32 | 55.41 | 69.26 | 80.91 | 95.30 | 87.52 | 98.28 | 89.26 | 93.55 | 82.46 |
| ETSformer | (2022) | 87.44 | 79.23 | 83.13 | 85.13 | 84.93 | 85.03 | 92.25 | 55.75 | 69.50 | 90.02 | 80.36 | 84.91 | 99.31 | 85.28 | 91.76 | 82.87 |
| LightTS | (2022a) | 87.10 | 78.42 | 82.53 | 82.40 | 75.78 | 78.95 | 92.58 | 55.27 | 69.21 | 91.98 | 94.72 | 93.33 | 98.37 | 95.97 | 97.15 | 84.23 |
| FEDformer | (2022b) | 87.95 | 82.39 | 85.08 | 77.14 | 80.07 | 78.57 | 90.47 | 58.10 | 70.76 | 90.17 | 96.42 | 93.19 | 97.31 | 97.16 | 97.23 | 84.97 |
| TimesNet (I) | (2023) | 87.76 | 82.63 | 85.12 | 82.97 | 85.42 | 84.18 | 91.50 | 57.80 | 70.85 | 88.31 | 96.24 | 92.10 | 98.22 | 92.21 | 95.21 | 85.49 |
| TimesNet (R) | (2023) | 88.66 | 83.14 | 85.81 | 83.92 | 86.42 | 85.15 | 92.52 | 58.29 | 71.52 | 86.76 | 97.32 | 91.74 | 98.19 | 96.76 | 97.47 | 86.34 |
| CrossFormer | (2023) | 83.6 | 76.61 | 79.70 | 84.68 | 83.71 | 84.19 | 92.04 | 55.37 | 69.14 | 88.49 | 93.48 | 90.92 | 97.16 | 89.73 | 93.30 | 83.45 |
| PatchTST | (2023) | 87.42 | 81.65 | 84.44 | 84.07 | 86.23 | 85.14 | 92.43 | 57.51 | 70.91 | 80.70 | 94.93 | 87.24 | 98.87 | 93.99 | 96.37 | 84.82 |
| ModernTCN | (2024) | 87.86 | 83.85 | 85.81 | 83.94 | 85.93 | 84.92 | 93.17 | 57.69 | 71.26 | 91.83 | 95.98 | 93.86 | 98.09 | 96.38 | 97.23 | 86.62 |
| LETO | (ours) | 88.20 | 85.52 | **86.84** | 83.50 | **89.27** | **86.29** | **93.20** | 57.10 | 70.81 | **92.00** | 96.73 | **94.31** | 99.20 | 94.61 | 96.85 | **87.02** |

where $\mathcal{M}_0^{(1)}$ is the initial state of the memory, which we let $\mathcal{M}_0^{(1)} = \mathbf{I}$ for the simplicity. Replacing this gradient in equation Variant 2, we have:

$$\mathcal{M}_{t,v}^{(1)} = \alpha_{t,v}\mathcal{M}_{t-1,v}^{(1)} - \eta_{t,v}\left(\underbrace{(\boldsymbol{k}_t - \boldsymbol{v}_t)}_{\mathbf{u}_t}\boldsymbol{k}_t^\top\right) + \beta_{t,v}\mathcal{M}_{t-1,v}^{(2)} - \gamma_{t,v}\left(\mathcal{M}_t^{(2)}\boldsymbol{k}_t\boldsymbol{k}_t^\top - \boldsymbol{v}_t\boldsymbol{k}_t^\top\right), \quad (20)$$

where we let $\eta_{t,v} = \gamma_{t,v} = 1$. Also, for the attention module, we use polynomials with degree 1 to approximate the softmax attention (which is the special instance and the weaker version of our design, i.e., considering only the first two terms of the Taylor series). The resulting formula can be written as:

$$\mathcal{M}_{t,v}^{(1)} = \alpha_{t,v}\mathcal{M}_{t-1,v}^{(1)} - \eta_{t,v}\mathbf{u}_t\boldsymbol{k}_t^\top + \beta_{t,v}\mathcal{M}_{t-1,v}^{(2)} - \gamma_{t,v}\mathcal{M}_t^{(2)} + \gamma_{t,v}\mathbf{u}_t\boldsymbol{k}_t^\top, \quad (21)$$

which is equivalent to the 2-dimensional linear recurrence with diagonal transition matrix. Therefore, as proven by Baron et al. (2024), the recurrence can model full-rank matrix.

On the other hand, the univariate version of this recurrence (i.e., $\gamma_{t,v} = 0$) results in linear attention formulation, which is limited and cannot express full-rank matrices.

Table 15: Full results for the classification task (accuracy %). We omit "former" from the names of Transformer-based methods. For all methods, the standard deviation is less than 0.1%. A higher average accuracy indicates a better prediction. Best performance is highlighted in **red**, and the second-best is underlined. We take the average of 5 separate runs for each dataset.

| Datasets / Models | LSTM (1997a) | LSTNet (2018) | LSSL | Trans. (2017) | Re. (2020) | In. (2021) | Pyra. (2021) | Auto. (2021) | Station. (2022b) | FED. (2022b) | /ETS. (2022) | /Flow. (2022b) | /DLinear (2023a) | LightTS. (2022a) | /TimesNet (2023) | /PatchTST (2023) | /MTCN (2024) | **LETO** (ours) |
|---|---|---|---|---|---|---|---|---|---|---|---|---|---|---|---|---|---|---|
| EthanolConcentration | 32.3 | 39.9 | 31.1 | 32.7 | 31.9 | 31.6 | 30.8 | 31.6 | 32.7 | 31.2 | 28.1 | 33.8 | 32.6 | 29.7 | 35.7 | 32.8 | 36.3 | **38.8** |
| FaceDetection | 57.7 | 65.7 | 66.7 | 67.3 | 68.6 | 67.0 | 65.7 | 68.4 | 68.0 | 66.0 | 66.3 | 67.6 | 68.0 | 67.5 | 68.6 | 68.3 | 70.8 | **71.3** |
| Handwriting | 15.2 | 25.8 | 24.6 | 32.0 | 27.4 | 32.8 | 29.4 | 36.7 | 31.6 | 28.0 | 32.5 | 33.8 | 27.0 | 26.1 | 32.1 | 29.6 | 30.6 | **32.9** |
| Heartbeat | 72.2 | 77.1 | 72.7 | 76.1 | 77.1 | 80.5 | 75.6 | 74.6 | 73.7 | 73.7 | 71.2 | 77.6 | 75.1 | 75.1 | 78.0 | 74.9 | 77.2 | **78.3** |
| JapaneseVowels | 79.7 | 98.1 | 98.4 | 98.7 | 97.8 | 98.9 | 98.4 | 96.2 | 99.2 | 98.4 | 95.9 | 98.9 | 96.2 | 96.2 | 98.4 | 97.5 | **98.8** | 98.5 |
| PEMS-SF | 39.9 | 86.7 | 86.1 | 82.1 | 82.7 | 81.5 | 83.2 | 82.7 | 87.3 | 80.9 | 86.0 | 83.8 | 75.1 | 88.4 | 89.6 | 89.3 | 89.1 | **89.6** |
| SelfRegulationSCP1 | 68.9 | 84.0 | 90.8 | 92.2 | 90.4 | 90.1 | 88.1 | 84.0 | 89.4 | 88.7 | 89.6 | 92.5 | 87.3 | 89.8 | 91.8 | 90.7 | 93.4 | **94.4** |
| SelfRegulationSCP2 | 46.6 | 52.8 | 52.2 | 53.9 | 56.7 | 53.3 | 53.3 | 50.6 | 57.2 | 54.4 | 55.0 | 56.1 | 50.5 | 51.1 | 57.2 | 57.8 | 60.3 | **61.1** |
| SpokenArabicDigits | 31.9 | 100.0 | 100.0 | 98.4 | 97.0 | 100.0 | 99.6 | 100.0 | 100.0 | 100.0 | 100.0 | 98.8 | 81.4 | **100.0** | 99.0 | 98.3 | 98.7 | 98.7 |
| UWaveGestureLibrary | 41.2 | 87.8 | 85.9 | 85.6 | 85.6 | 85.6 | 83.4 | 85.9 | 87.5 | 85.3 | 85.0 | 86.6 | 82.1 | 80.3 | 85.3 | 85.8 | 86.7 | **87.1** |
| Average Accuracy | 48.6 | 71.8 | 70.9 | 71.9 | 71.5 | 72.1 | 70.8 | 71.1 | 72.7 | 70.7 | 71.0 | 73.0 | 67.5 | 70.4 | 73.6 | 72.5 | 74.2 | **75.07** |

As an important note: Theorem 1 is intentionally stated in a linear recurrent setting and should be read as a structural comparison within a *linearized* model class rather than a full theory of the nonlinear LETO architecture. Recall from equation 11 that our cross-variate block uses a kernelized attention with an explicit feature map $\phi$, so that in the feature space induced by $\phi$ the memory $\mathcal{M}_{t,v}^{(2)}$ evolves linearly in $\phi(\hat{\boldsymbol{k}}_{t,i})$. When we fix $\phi$ and consider the joint update of $\mathcal{M}^{(1)}$ and $\mathcal{M}^{(2)}$ in equation Variant 2, the resulting dynamics fall exactly into the class of interconnected linear recurrences analyzed in Theorem 1. Within this linearized class, the theorem shows that coupling the two memories allows us to realize full-rank kernels with $\mathcal{O}(1)$ parameters, whereas using two independent modules of the form equation Variant 1 requires at least $\mathcal{O}(N)$ parameters to represent rank-$N$ interactions. We do *not* claim that this result characterizes the full nonlinear LETO or provides general expressivity bounds beyond rank; rather, it offers mechanistic intuition about the parameter efficiency of the inter-connected design. This intuition is complemented by our empirical ablations (Table 4 and Appendix E.2), where removing the cross-variate memory, removing the linearized attention, or decoupling the two memories all lead to consistent performance degradation compared to the full LETO.

# J VISUALIZATIONS

## J.1 LONG TERM FORECASTING

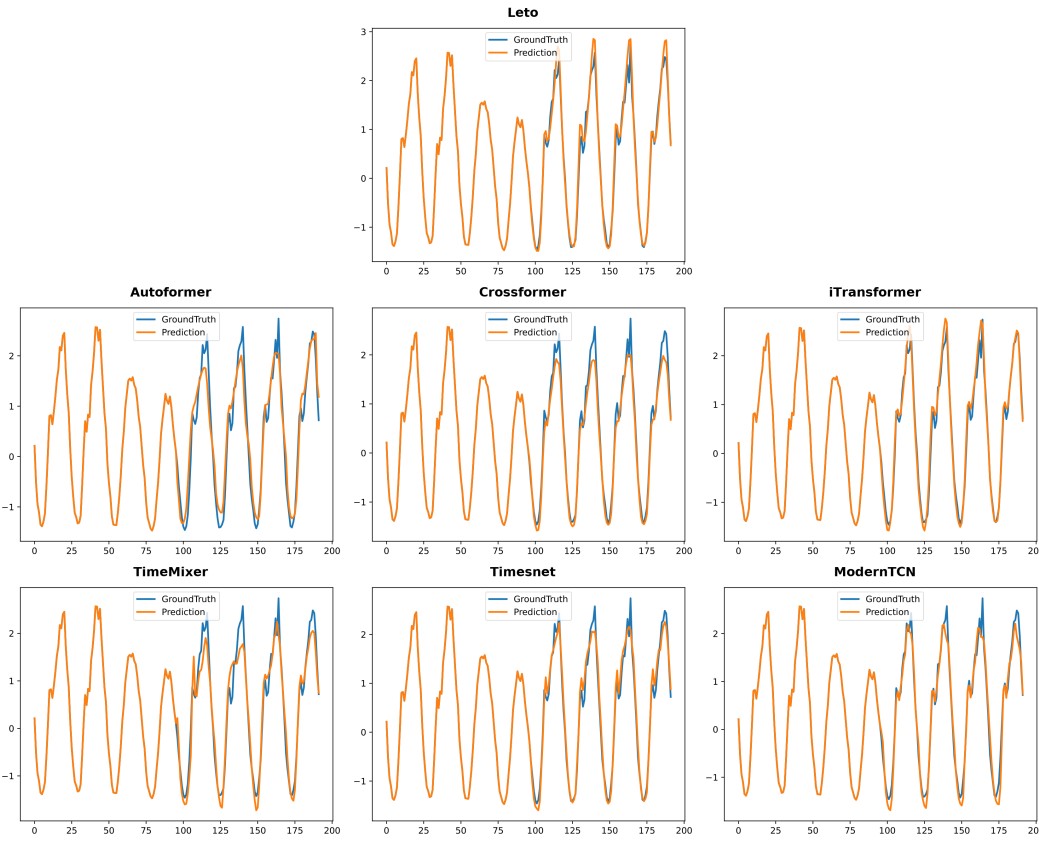

Figure 3: Visualization of Traffic Long Term Forecasting results given by models under the input-96-predict-96 setting. The blue lines stand for the ground truth and the orange lines stand for predicted values.

## J.2 ULTRA LONG TERM FORECASTING

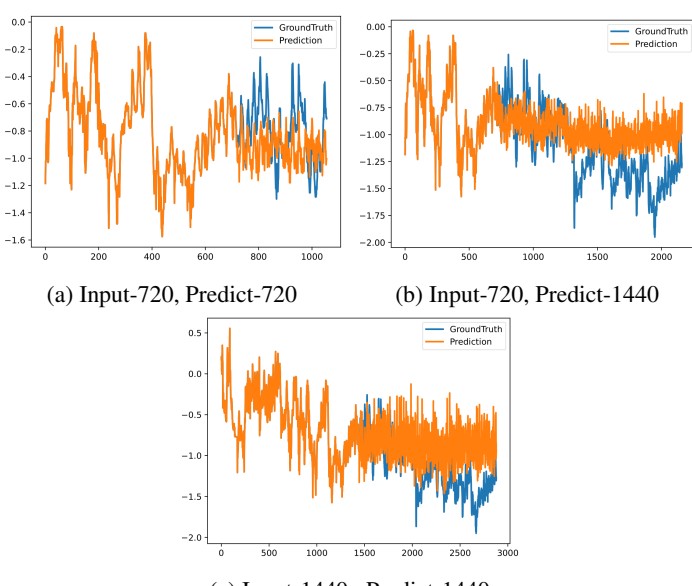

(a) Input-720, Predict-720      (b) Input-720, Predict-1440

(c) Input-1440 –Predict-1440

Figure 4: Ultra-long-horizon forecasting examples on the ETTh1 dataset. The blue lines stand for the ground truth and the orange lines stand for predicted values.

