# OpenReview forum: "Leto: Modeling Multivariate Time Series with Memorizing at Test Time"
_ICLR.cc/2026/Conference — Submitted to ICLR 2026_

### Official Review · Reviewer_wNzq · 2025-10-29

**Soundness:** 2
**Presentation:** 3
**Contribution:** 2
**Rating:** 4
**Confidence:** 5

**Summary:**

The paper introduces LETO, a 2D meta in-context memory architecture for multivariate time series modeling. Unlike traditional sequence models that process each variable independently or ignore inter-variate dependencies, LETO maintains temporal inductive bias while being permutation equivariant across variates. The core idea is to use interconnected memory modules that capture both temporal dynamics and cross-variable interactions through a shared meta-memory mechanism. Theoretical analysis (Theorem 1) suggests that such interconnected modules can represent higher-rank kernels more efficiently than independent ones. Extensive experiments on forecasting, classification, and anomaly detection benchmarks show that LETO achieves state-of-the-art performance across short-, long-, and ultra-long-term horizons.

**Strengths:**

The paper is generally well-organized and easy to follow, and it provides comprehensive experimental validation demonstrating the effectiveness of the proposed 2D meta-memory architecture.

**Weaknesses:**

1) The motivation for LETO is unclear. The paper asserts that existing multivariate time series models lack temporal inductive bias and inter-variate dependency modeling, yet prior architectures (e.g., TCNs, Transformers with positional encodings, TimesNet, Graph-based models) already incorporate both in different ways. The authors should clarify (1) what exact deficiency remains in these approaches, and (2) how LETO’s 2D memory introduces a distinct, structural bias rather than another learned mechanism.

2) Theorem 1 provides only a linear-algebraic argument suggesting that interconnected memory modules can represent higher-rank kernels with fewer parameters. However, this result holds only under linear recurrent assumptions and does not reflect the nonlinear, attention-based nature of LETO’s actual implementation. Moreover, the proof does not present an explicit constructive mapping or quantify expressivity beyond rank. Consequently, while the theorem offers an intuition for parameter efficiency, it does not constitute a rigorous theoretical justification of the proposed architecture’s behavior.

3) LETO’s meta-memory module may implicitly retain information beyond the fixed input window used for competing models. While the paper claims identical input–output settings across baselines, the presence of a persistent memory state effectively extends the model’s historical context. This raises fairness concerns, as LETO may have access to longer-term information than baselines restricted to feedforward input windows. The authors should clarify whether the meta-memory is reset between batches and whether the effective receptive field is comparable across models.

4) Although the paper benchmarks against a wide range of prior models, it does not provide diagnostic experiments or analyses that substantiate the claimed limitations of existing methods. The argument that previous architectures lack temporal inductive bias or inter-variate dependency modeling remains qualitative. There is no evidence demonstrating that such deficiencies are the cause of baseline underperformance. As a result, the experiments validate LETO’s performance but not its motivation.

5) The paper repeatedly refers to temporal “dynamics” and “state evolution,” suggesting an underlying dynamical-systems perspective. However, the theoretical section provides only a static, linear-rank argument and does not model or analyze any stochastic or continuous-time process. Is LETO intended to approximate an underlying dynamical system (e.g., through an implicit SDE/ODE formulation), or is the term “dynamics” purely metaphorical? If the former, could the authors formalize how the meta-memory update corresponds to a discrete approximation of such dynamics, and discuss its stability or stochastic robustness properties?


6) The paper contains several minor grammatical and formatting inconsistencies (e.g., duplicated words such as “the the,”in the abstract section, inconsistent use of “includes,” subject–verb mismatches, and irregular equation referencing). Terminology is occasionally inconsistent (“meta in-context memory,” “meta-memory,” etc.), and some sentences are overly long or rhetorically strong. A careful language and formatting revision would improve readability and professionalism.

**Questions:**

1) Existing multivariate time series models such as TCNs, TimesNet, and Transformer variants already incorporate both temporal inductive bias (via convolutions or positional encoding) and inter-variate modeling (via attention or graph structures). What specific structural deficiency do these methods still have that LETO’s 2D memory addresses? Please clarify how LETO’s “meta in-context memory” introduces a genuinely new inductive bias rather than another learned attention mechanism.

2) Theorem 1 provides a linear-rank argument under linear recurrence assumptions, but LETO is nonlinear and attention-based. How does this result theoretically connect to the actual architecture? Could the authors formalize or empirically validate the claimed expressivity or parameter-efficiency advantage beyond this simplified setting?

3) If the model indeed embodies implicit dynamics, could the authors clarify how its update rule relates to a discrete or stochastic dynamical formulation and whether any stability guarantees hold?

---

> ### Author Response · Authors · 2025-11-22
>
> We thank the reviewer for the careful reading and thoughtful comments. Below we clarify (1) the structural motivation and novelty of LETO, (2) how Theorem 1 connects to the implemented architecture, (3) fairness concerns around meta-memory and context length, (4) the role of diagnostic experiments and related work, and (5) our use of “dynamics” language. We also commit to specific edits to improve clarity and presentation.
>
> ## Weakness 1
>
> > The motivation for LETO is unclear.
>
> **Response** We agree that recent multivariate time-series models (TCNs, TimesNet, Transformer variants with positional encodings, graph-based models) do indeed incorporate both temporal structure and cross-variate interactions. We want to strongly emphasize that our claim is not that they completely lack these ingredients, but that they typically realize them in a factorized or asymmetric way:
>
> - Many architectures adopt a strong temporal mechanism (RNN/SSM/TCN) but treat variates in a fixed order, so their cross-variate modeling is order-sensitive and not permutation-equivariant.
> - Others emphasize permutation-equivariant mixing across variates (e.g., self-attention over channels, graph modules) but apply this in a largely feedforward or shallow manner over time, rather than via a recurrent memory that explicitly evolves.
>
> This is structurally different from stacking a temporal block followed by a variate block (or vice versa) that only communicate via concatenation. Ablations in Table 4 already show that removing the inter-connection (e.g., decoupled variant, or dropping cross-variate attention) leads to a clear and consistent drop in performance across datasets. We will be sure to clarify in Sec. 3 that the novelty lies in this 2D meta in-context memory with permutation-equivariant variate update and recurrent temporal update that are tightly coupled, rather than in using attention per se.
>
> LETO’s inductive bias is that both temporal and variate dimensions are modeled by inter-connected memories:
> (i) a recurrent time memory with gating/retention that captures temporal dynamics, and
> (ii) a permutation-equivariant cross-variate memory, and crucially
> (iii) these two memories are updated jointly at every step (Variant 2), so each time update conditions on a variate-equivariant summary and each variate update conditions on the time state.
>
>
> ## Weakness 2/Question 2
>
> >Connection between Theorem 1 and the nonlinear attention-based implementation.
>
> **Response** We appreciate the concern that Theorem 1 is stated in a linear recurrent setting, while LETO is instantiated with nonlinear/attention-based modules. Our intention is to use Theorem 1 as a mechanistic justification of why inter-connecting two memories can yield higher-rank effective kernels with fewer parameters than independent blocks, not as a full theory of the nonlinear network.
>
> Concretely:
>
> - The cross-variate module in LETO uses a linear-attention style update with an explicit feature map $\phi$ for the softmax kernel. In this feature space, the attention operation is linear in $\phi(X)$.
> - Under this linearization, the combined “time + variate” memory update fits into the form analyzed in Theorem 1: we have two linear recurrent operators acting on (possibly different) feature representations that are inter-connected. The theorem then formalizes that such inter-connection yields higher-rank kernels than independent memories at the same parameter budget.
>
> We fully agree that this does not yet characterize the full nonlinear network, nor does it prove general expressivity bounds. In the revision we will add a remark directly after Theorem 1 stating that it applies to the linearized variant (time recurrence + linearized cross-variate attention) and carefully scope what is, and is not, claimed. We will explicitly write the feature-map form of our linear attention in Sec. 3 and show how the resulting block matches the assumptions of Theorem 1.
>
> We view this as giving principled intuition about parameter efficiency of the inter-connected design, which is then supported empirically by the ablations (“w/o linear attention”, “w/o cross-variate attention”, “decoupled memories”) that show consistent degradation compared to full LETO.

---

> ### Author Response · Authors · 2025-11-22
>
> ## Weakness 3
>
> >Meta-memory and effective context length / fairness.
>
> **Response** We apologize for not stating this more explicitly. In all reported experiments: The meta-memory is re-initialized for each input sequence at both training and evaluation. No state is carried over across batches or examples. For long sequences that are processed in chunks for efficiency, we respect the configured look-back window and reset at sequence boundaries; there is no leakage of information across different sequences.
> Consequently, LETO never accumulates information beyond the fixed input window used for the baselines; its effective receptive field matches the stated context length. Meta-learning only affects the initialization of the memory state (so it can adapt quickly within the window), not a persistent carry-over between sequences.
> We will clarify in text that memory states are reset per sequence and that input–output settings and receptive fields are matched across baselines and explicitly state in that no cross-batch or cross-sequence state is preserved.
>
> ## Weakness 4/Question 1
>
> >Baseline limitations and diagnostic experiments.
>
> **Response** We agree with the reviewer that, ideally, we would like to reference a full causal study that isolates why each baseline underperforms (e.g., by systematically breaking temporal inductive bias, cross-variate coupling, or permutation equivariance). Performing such a study across the many model classes we cover (Transformers, TCN, TimesNet, SSMs, hybrids) would require substantial experimentation and is beyond the scope of this paper.
> Instead, our approach is:
> - Empirical validation: we compare LETO extensively against a wide range of baselines across forecasting, classification, and anomaly detection, and observe consistent gains, especially on long/ultra-long horizons.
> - Targeted ablations: within LETO, we isolate the impact of time memory, cross-variate attention, linear attention, and the coupled vs. decoupled design (Table 4), which directly supports the architectural choices.
> - Connection to existing analyses: several recent works explicitly report that standard Transformer-style MTS models struggle to jointly capture temporal and variate dependencies or lack temporal inductive bias even when they employ attention. For discussions on limitations of Transformer based architectures in time series see: [1,2,3,4,5], for discussions on limitations of TimesNet see [6], and discussions on limitations of TCN see [7].
> We will (i) soften the rhetoric around “lack” of temporal inductive bias/inter-variate modeling and instead phrase it as “limitations documented in prior work."
>
> ## Weakness 5/Question 3
>
> >Use of “dynamics” and dynamical-systems interpretation.
>
> **Response** Our use of “dynamics” refers to the discrete-time evolution of the time memory, not to an explicit continuous-time SDE/ODE model. The time memory follows a gated linear update of the form:
> $h_{t+1} = \alpha_t \odot h_t + \eta_t \odot f(x_t, m_t)$
> where $\alpha_t \in [0,1]$ and $\eta_t$ are learned gates and $m_t$ is the cross-variate summary. This is a linear time-varying system driven by a bounded input $f(x_t, m_t)$ (bounded by activations and normalization). Under standard conditions on the gates (e.g., $\|\alpha_t\|_\infty \leq 1 - \epsilon$), the system is contractive and the memory remains bounded. We will also clarify in Sec. 3.2 that we use “dynamics” in this discrete-time sense and do not claim to recover an underlying physical SDE/ODE; instead, we adopt dynamical language to emphasize the stateful nature of the time memory.
>
> ## Weakness 6
>
> >Writing/formatting
>
> **Response** We appreciate the detailed pointers. We will carefully correct duplicated words, equation references, and other minor grammatical issues, and we will standardize terminology to “meta in-context memory (meta-memory)” throughout to avoid confusion.
>
> Please let us know if you have further questions; we are very open to discussion. Thank you again for your time and feedback!
>
>
> - [1] UniTST: Effectively Modeling Inter-Series and
> Intra-Series Dependencies for Multivariate Time Series Forecasting https://arxiv.org/pdf/2406.04975
> - [2] HYDRA: Dual Exponentiated Memory for Multivariate Time Series Analysis https://arxiv.org/pdf/2511.00989
> - [3] Why Do Transformers Fail to Forecast Time Series In-Context?https://arxiv.org/pdf/2510.09776
> - [4] Context Neural Networks: A Scalable Multivariate Model for Time Series Forecasting https://arxiv.org/abs/2405.07117
> - [5] A Closer Look at Transformers for Time Series Forecasting: Understanding Why They Work and Where They Struggle: https://openreview.net/pdf?id=kHEVCfES4Q
> - [6] INVNET Inverted TimesNet for Multivariate Time Series Forecasting https://ieeexplore.ieee.org/document/10936236
> - [7] ModernTCN Revisited: A Critical Look at the Experimental
> Setup in General Time Series Analysis https://openreview.net/pdf?id=R20kKdWmVZ

---

### Official Review · Reviewer_j6VP · 2025-10-31

**Soundness:** 2
**Presentation:** 1
**Contribution:** 1
**Rating:** 2
**Confidence:** 4

**Summary:**

This paper presents an architecture, LETO, for multivariate time series analysis. The authors frame this as a "native 2D" model that is derived from the principles of meta-learning and memorizing at test time (TTM). The model attempts to properly handle the distinct properties of the time dimension (requiring temporal inductive bias) and the variate dimension (requiring permutation equivariance). While the initial perspective is interesting, the paper's central claims (regarding its "native 2D" nature, TTM/meta-learning foundation, and novelty) do not sufficiently align with its actual contributions.

**Strengths:**

1. **Interesting Conceptualization:** The paper correctly identifies a key challenge in multivariate time series: the temporal dimension requires a causal, sequential model (like an RNN), while the variate dimension requires a permutation-equivariant model (like an Attention mechanism). The proposal to build a hybrid "native 2D" architecture to address this is an interesting perspective.
2. **Novel Technical Component:** The technical solution to merge the RNN-like state with the attention-like mechanism, by using a kernelized attention approximated by its Taylor series, is a clever technical contribution.
3. **Extensive Experiments:** The model is evaluated across a wide range of tasks, including long-term and short-term forecasting, classification, and anomaly detection, demonstrating extensive empirical effort.

**Weaknesses:**

The paper suffers from significant weaknesses in its theoretical framing and the completeness of its comparisons.
1. **Hard to Follow:**
   * Overall, the paper is difficult to read, lacks a clear structural diagram, and Figure 1 does not effectively and clearly illustrate the structure of the proposed model.
   * Figure 2 is too small, making it very difficult to read. While we understand this may be due to page limits, we strongly suggest replacing it with a clearer, larger figure in any revision.
   * The Appendix numbering is flawed: it jumps from Table 5 to Table 7, with Table 6 missing.
   * Table 7 is present in the Appendix but appears to be unreferenced in the text, leaving its context and purpose unclear.
2. **Mismatched Theoretical Framing (TTM/Meta-Learning):** The paper's claim on "meta-learning" and "memorizing at test time" as motivation for its architecture is unconvincing. By the paper's definition, any recurrent model with a state update (RNN, LSTM, or even modern SSMs) could be reframed as a "test-time learner" that adapts its internal "memory" to a new input. This is likely why the community does not generally apply this terminology to state-based sequence models, as it does not provide functional novelty or a clear distinction from standard recurrent processing. Unless the authors can justify the necessity of using the TTM/Meta-learning terminology and framework, or refactor the paper's title and narrative to de-emphasize TTM/Meta-learning and focus the contribution on a native 2D multi-variate temporal attention mechanism, a recommendation for acceptance will not be given. This is because the TTM/Meta-learning framing is not just mismatched but potentially misleading.
3. **Lack of Analytical Comparison (vs. iTransformer)**: The paper includes iTransformer in its experiments, acknowledging it as a baseline. However, the core idea of LETO's variate-handling mechanism (using an attention-like module for permutation equivariance across channels) is conceptually very similar to iTransformer's core idea (inverting the Transformer to apply attention across variates to model correlations). Given this strong conceptual overlap, the paper is missing a crucial analytical or theoretical discussion differentiating LETO's hybrid approach from iTransformer's pure-attention approach.
4. **Conditional Omission of SSM Baselines (Mamba):** The paper omits all modern SSMs (like Mamba) from its tables. We note the justification in Appendix D, which states that other literature (ModernTCN) has shown superiority over SSMs, and thus comparing to ModernTCN is sufficient. This reasoning is acceptable only if the authors confirm that the experimental setup (hyperparameters, training, testing, etc.) for ModernTCN used in this paper is generally identical to the setup in the paper of ModernTCN. Without this confirmation, the comparison is not valid. Even so, given that LETO is a recurrent-style model, a direct comparison to at least one modern SSM baseline is still encouraged for a more complete and convincing evaluation.
5. **Lack of Critical Ablation Studies:** The paper's core technical "trick" is the use of a Taylor series to approximate the softmax kernel. The choice of a 3rd-order approximation appears arbitrary. There is no ablation study on the order of this approximation (e.g., 1st vs. 2nd vs. 3rd order) to justify this key design choice. This makes the "principled derivation" from the TTM framework seem more like an ad-hoc engineering decision.

**Questions:**

See the Weaknesses.

---

> ### Author Response · Authors · 2025-11-22
>
> Thank you for your time and valuable comments. We really appreciate it. We believe that there might have been some misunderstanding about our contribution and paper and hope that the following responses fully address your concerns. We address each weakness and concern below.
>
> ## Weakness 1:
>
> > Figure 2 is too small, making it very difficult to read.
>
> **Response:** Thank you for bringing this to our attention. Following your suggestion, we have changed the figure and made its font much bigger, close to the main part of the paper.
>
> > Table 7 is present in the Appendix but appears to be unreferenced in the text.
>
> **Response:** Thank you for bringing this to our attention. Following your suggestion, we have revised the paper and properly reference Table 7 in the main text.
>
> ## Weakness 2:
>
> > Mismatched Theoretical Framing (TTM/Meta-Learning) ... any recurrent model with a state update (RNN, LSTM, or even modern SSMs) could be reframed as a "test-time learner" that adapts its internal "memory" to a new input.
>
> **Response:** Thank you for mentioning this; it is a great question. Unfortunately, there is a misunderstanding in the community, even among experts, around this new research field of test time training/memorization and its corresponding representation of recurrent models. We clarify that how this formulation is different from common formulation of recurrent models:
> - Most modern recurrent models (or maybe even all), can be reformulated as a bi-level process where the memory updates its own weights in the inner-loop and all other parameters are updated in the outer-loop. Despite this reformulation, models such as linear attention, modern SSMs, etc., are not performing meta-learning or full test time memorization, mainly due to the fact that their initial state of the memory starts from zero. Therefore, these models only learn how to properly project the input into keys, values, and queries, and then update their memory based on the learning rule they are defined based on. There is no process of learning to learn at test time.
> - On the other hand, a set of new models such as TTT [1], Titans[2], and our recurrent memory in Leto, are based on a meta-learning process, where we meta-learn the initial value of the memory so it can adapt fast to the new context or a new time series.
> - The closed-form recurrent formulation of RNNs, linear attentions, and SSMs, limits their state or memory to be linear. That is, memory $M_t$ is either a matrix or a vector, and it is not clear that how one can go beyond a matrix and use a neural network or even a tensor as the state or memory of these recurrent models. The more expressive state is a critical step for improving RNNs as it can enahnce the capacity of the memory. The TTM/Meta-learning formulation allows us to define more powerful hidden states (or memory) but the resulting model does not have a closed-recurrent form.
> - In time series data, specifcally, when we are dealing with short context window as compared to language tasks (this setup is common in time series, as even ultra long forecasting tasks uses only less than 1K tokens), the meta-learning component is a very important part since it helps the model to meta-learn an initial state of memory so it can adapt to the new time series fast.
>
> We hope that our above response has clarified the distinction of meta-learning and closed-form recurrent formulations of RNNs. We would be happy to clarify any remaining concern around this topic. We believe the importance of our design for multivariate time series data and specifically the native 2D design (as also mentioned by the reviewer) is a valuable contribution and can be a timely and useful contribution for the community to build on.
>
> However, while we believe that the initial title is a good representative of our approach (due to meta-learning of initial state), after our above explanations, if the reviewer believes that changing the title and removing test time memorization can result in their support of acceptance, we would be very happy to change the title.

---

> > ### Author Response · Authors · 2025-11-22
> >
> > ## Weakness 3
> >
> > Lack of Analytical Comparison (vs. iTransformer)
> >
> > **Response:** Please note that one of the main contribution of the paper is to advocate for the native 2D design for the multivariate time series data. As it is shown in Figure 1, a multivariate time series data has an underlying 2D dependencies between the states. Please see our response to Reviewer wNzq providing several recent works which discuss the shortcomings of pure attention/transformer based frameworks for time series forecasting. We reference the names of the papers as they appear in our reference list to Reviewer wNzq. UniTST shows that existing Transformer models that separately apply time-wise and variate-wise attention (including iTransformer-style designs) still cannot directly model cross-time cross-variate dependencies, and that such channel-mixing Transformers can even be outperformed by simple linear baselines on standard benchmarks. Hydra further argues that pure Transformers suffer from quadratic time and memory complexity in the context length, making them difficult to scale to long horizons, and that attention-based mixing over noisy variates can hurt robustness, motivating more structured memory-based designs. In parallel, “A Closer Look at Transformers for Time Series Forecasting” finds that even architectures with explicit inter-variate attention (e.g., iTransformer) mainly rely on intra-variate patterns, with inter-variate attention contributing relatively little, and that their performance is highly sensitive to simple preprocessing (e.g., Z-score normalization) and can degrade under non-stationary settings. We hope our current and extended ablation studies and our robust performance compared to iTransformer makes a strong case for the native 2D design.
> >
> > ## Weakness 4
> >
> > >Conditional Omission of SSM Baselines (Mamba)
> >
> > **Response:** We kindly want to bring to your consideration that:
> > - We have already included a SSM and Mamba-based baseline in our experiments. Specifically, we already have included Simba [3] model in our experiments as a baseline, which is a Mamba-based model that is specifically designed for multivariate time series forecasting. We demonstrate that our model, with statistical significance, performs better across different tasks. To further address your comment, we have included the results for Time Machine [4] and MambaMixer [5], which are recent Mamba-based models applied to multivariate time series forecasting. On nearly all long term multivariate forecasting tasks, Leto outperforms these Mamba-based models.
> >
> > On the long-term forecasting benchmark (average over horizons $\{96, 192, 336, 720\}$). Results are as follows:
> >
> > | Dataset | LETO (MSE / MAE) | Time Machine (MSE / MAE) | Mamba Mixer (MSE / MAE) |
> > |--------|-------------------|---------------------------|--------------------------|
> > | ETTm1  | **0.347 / 0.375** | 0.375 / 0.392             | 0.361 / 0.389            |
> > | ETTm2  | **0.249 / 0.302** | 0.268 / 0.318             | 0.267 / 0.310            |
> > | ETTh1  | **0.393 / 0.401** | 0.417 / 0.419             | 0.403 / 0.408            |
> > | ETTh2  | **0.318 / 0.381** | 0.344 / 0.382             | 0.333 / 0.385            |
> > | Weather| **0.216 / 0.253** | 0.243 / 0.273             | 0.239 / 0.261            |
> > | ECL    | **0.149 / 0.247** | 0.170 / 0.263             | 0.169 / 0.258            |
> >
> > Furthermore, we confirm that the experimental setup (hyperparameters, training, testing, etc.) for ModernTCN used in this paper is identical to the setup used in ModernTCN paper. Also, our reported results for ModernTCN faithfully replicate those reported in the original paper.

---

> ### Author Response · Authors · 2025-11-22
>
> ## Weakness 5
>
> > Lack of Critical Ablation Studies
>
> **Response:** We fully agree that the original ablations in Table 4 may seem high level and some design choices may be unclear. Regarding the choice of the third order Taylor series we provide an ablation study which provides strong evidence for its use. We refer the reviewer to our ablation studies on orders 1-4 for the Taylor series approximation of the softmax kernel, as well as additional ablations on chunk size $b$ in parallelizable training and $\lambda$ which multiplies the cross-variate term $M^{(2)}_{t,v}$ in our response to reviewer krLC. We hope that the ablation study on Taylor order makes clear the expressive nature of the third order Taylor approximation compared to other orders, and the other ablations validate the design choice of Leto.
>
> Thank you very much once again for your time and valuable feedback. We hope that our above responses have fully addressed your concerns. We would be happy to clarify or address any remaining concerns you might have.
>
> - [1] Learning to (Learn at Test Time): RNNs with Expressive Hidden States: https://arxiv.org/abs/2407.04620
> - [2] Titans: Learning to Memorize at Test Time: https://arxiv.org/abs/2501.00663
> - [3] SiMBA: Simplified Mamba-based Architecture for Vision and Multivariate Time series: https://arxiv.org/pdf/2403.15360
> - [4] TimeMachine: A Time Series is Worth 4 Mambas for Long-term Forecasting: https://arxiv.org/pdf/2403.09898
> - [5] MambaMixer: Efficient Selective State Space Models with Dual Token and Channel Selection: https://arxiv.org/abs/2403.19888
> - [6] iTransformer: Inverted Transformers Are Effective for Time Series Forecasting: https://arxiv.org/abs/2310.06625

---

### Official Review · Reviewer_krLC · 2025-11-06

**Soundness:** 2
**Presentation:** 2
**Contribution:** 2
**Rating:** 4
**Confidence:** 4

**Summary:**

This paper introduces LETO, a native 2-dimensional memory-based architecture for modeling multivariate time series. LETO fuses two meta in-context memory modules: a recurrent temporal memory for capturing time dynamics and a permutation-equivariant attention-based memory for mixing information across variates. The paper provides a theoretically motivated formulation, implementation details for efficient parallelizable training, and comprehensive empirical evaluations across datasets for forecasting, classification, and anomaly detection. Ablation studies are presented to justify design choices.

**Strengths:**

1、The paper tackles a significant and longstanding challenge in time series modeling: capturing both long-range temporal and cross-variate dependencies without sacrificing efficiency or robustness.

2、 There are quite a few nice illustrations.

3、 This work focuses on an important problem that could have real-world applications.

4、 The figures and tables used in this work are clear and easy to read.

**Weaknesses:**

1、As a paper submitted to ICLR 2026, the comparison baselines chosen by the authors are relatively outdated, lacking evaluations against the latest methods from 2025. This limitation undermines the credibility of the reported performance gains and raises concerns about the reliability of the experimental conclusions.

2、The Related Work section does not adequately cover recent advances in multivariate modeling. In particular, it lacks discussion of several representative studies published in 2024–2025 that have significantly advanced this area. As a result, the literature review appears incomplete and fails to clearly position the proposed work within the current state of research.

3、While high-level architectural design is summarized (see Figure 1 on Page 5), there is a lack of granular description regarding data flow, batching, and precise ordering of operations—especially important for readers seeking reproducibility or trying to adopt the approach in their own pipelines, especially when parallelism is highlighted as a main practical advantage.

4、While the reported results are competitive, the ablation study (Table 4, Page 8) is limited in terms of scope. The architectural variants removed (“cross-variate attention”, “linear attention”, “weighted gating”) are natural ablations, but ablations for Taylor expansion order, chunk size in parallelization, or the extent of cross-memory coupling are missing. This weakens the argument for the chosen design.

5、In the NIPS 2024 workshop[1], some researchers pointed out that current methods sometimes use the "drop-last" trick [2] to improve performance. Therefore, It is recommended that you clarify whether the "drop - last" operation [2] was used in your paper in the implementation details section of your paper for transparency.

[1] Fundamental limitations of foundational forecasting models: The need for multimodality and rigorous evaluation

[2] TFB: Towards Comprehensive and Fair Benchmarking of Time Series Forecasting Methods

If my problem is solved, I will improve my score!

**Questions:**

Please see the weaknesses！

---

> ### Author Response · Authors · 2025-11-22
>
> We thank the reviewer for the careful reading, the positive comments on the clarity of our figures/tables, and the constructive suggestions. Below we address each weakness and the question in turn.
>
> ## Weakness 1
>
> > Recent baselines
>
> **Response** Following your suggestion, we have added several recent baselines beyond those already in Table 3:
> TimeKAN (ICLR 2025) [1] and TimeFilter (ICML 2025) [2] TimePro (ICML 2025) [3].  We adopt their official public implementations and hyperparameter grids, keeping the preprocessing and evaluation protocol unchanged.
>
> On the long-term forecasting benchmark (average over horizons $\{96, 192, 336, 720\}$). Results are as follows:
>
> | Dataset | LETO (MSE / MAE) | TimeKAN (MSE / MAE) | TimeFilter (MSE / MAE) | TimePro (MSE / MAE) |
> |--------|-------------------|----------------------|------------------------|----------------------|
> | ETTm1  | **0.347 / 0.375** | 0.376 / 0.395        | 0.377 / 0.393          | 0.391 / 0.400        |
> | ETTm2  | **0.249 / 0.302** | 0.277 / 0.322        | 0.272 / 0.321          | 0.281 / 0.326        |
> | ETTh1  | **0.393 / 0.401** | 0.408 / 0.411        | 0.420 / 0.428          | 0.438 / 0.438        |
> | ETTh2  | **0.318 / 0.381** | 0.383 / 0.404        | 0.364 / 0.397          | 0.377 / 0.403        |
> | Weather| **0.216 / 0.253** | 0.242 / 0.272        | 0.239 / 0.269          | 0.251 / 0.276        |
> | ECL    | **0.149 / 0.247** | 0.197 / 0.286        | 0.158 / 0.256          | 0.169 / 0.262        |
>
> On Short Term Forecasting, we compare with ROSE (ICML 2025) [4].
>
> | Dataset  | LETO (SMAPE / MASE / OWA)              | ROSE (SMAPE / MASE / OWA)      |
> |----------|----------------------------------------|--------------------------------|
> | Yearly   | **13.183** / **2.941** / **0.754**     | 13.302 / 3.014 / 0.833         |
> | Quarterly| **9.953** / **1.150** / **0.851**      | 9.998 / 1.165 / 0.885          |
> | Monthly  | **12.517** / 0.935 / **0.853**         | 12.650 / **0.915** / 0.866     |
> | Others   | **4.583** / **2.797** / **0.9001**     | 4.668 / 3.126 / 1.020          |
>
> These results (averaged over 3 seeds) demonstrate the strong performance of our model with recent baselines.
>
> We like to clarify that the baselines in the original Table 3 are not outdated. They include universally adopted and competitive models such as iTransformer, ModernTCN, TimeMixer, TimesNet, PatchTST. These methods continue to be used as strong baselines in recent benchmark papers and large-scale empirical studies. Our new results compared with new baselines reinforce that LETO remains competitive even when compared to the most recent non-foundation models.
>
> We also kindly point out that many of the state-of-the-art methods from 2025 are time series foundation models (TSFMs), pre-trained on massive heterogeneous corpora and evaluated primarily in zero-shot or few-shot regimes. Our setting is the standard supervised per-dataset forecasting regime with no cross-dataset pretraining and with parameter counts comparable to ModernTCN/iTransformer. Directly comparing LETO to TSFMs would conflate (i) architecture, (ii) model scale, and (iii) pretraining data. We propose a new architecture that can be used at the same scale as existing supervised models, and could in principle also be used as the backbone within a TSFM in future work.
>
> If the reviewer has specific non-foundation architectures from 2025 they would particularly like to see, we are happy to include them.
>
> ## Weakness 2
>
> > Related Work Discussion (Weakness 2)
>
> We agree that the Related Work can more clearly situate our work within the most recent literature. In the revised version we will explicitly discuss:
>
> - Recent multivariate forecasters based on advanced patching / hierarchical designs (ie TimeKAN), and how they primarily operate along the temporal axis with channel mixing layers, but still could benefit from an explicit coupled 2D memory view where temporal and variate memories are updated jointly at each step.
> - Recent TSFM and TSFM-benchmark works (e.g., large decoder-only forecasters and their zero-shot performance), clarifying that they study scaling and pretraining, whereas LETO targets architectural inductive bias at a fixed scale.
> - Recent analyses emphasizing the importance of fair evaluation and diagnostic experiments (e.g., works that expose drop-last and other pitfalls), and how we align our protocol with these recommendations (see “Drop-last” below).
>
> We will explicitly contrast LETO’s native 2D meta-memory with: (i) architectures that invert dimensions or patch along time but still rely on a single memory, and (ii) attention mixers that do not maintain a persistent variate memory state. See our response to Reviewer wNzq for more clarity on where our model sits in the literature.

---

> ### Author Response · Authors · 2025-11-22
>
> ## Weakness 3
>
> > Implementation Details and Parallelization (Weakness 3)
>
> We appreciate the request for more granular implementation details, especially given that parallelizable training is a key practical contribution. Our current draft sketches the idea in Sec. 3.3 and Appendix C; here we make the data flow and batching explicit. Concretely, the core training loop for LETO (in the forecasting setting) operates as follows.
>
> The input to the model is a tensor $X \in \mathbb{R}^{B \times T \times V \times d}$, where $B$ is the batch size, $T$ is the lookback length, $V$ is the number of variates, and $d$ is the feature dimension after embedding and normalization.
>
> For each time step $t$, we form variate-wise queries, keys, and values $Q\_t, K\_t, V\_t \in \mathbb{R}^{B \times V \times d}$. We then apply a Taylor-approximated softmax attention across the variate dimension (of size $V$) to obtain the cross-variate memory state $M^{(2)}\_t \in \mathbb{R}^{B \times V \times d}$. These states $M^{(2)}\_t$ are computed for all $t = 1, \dots, T$ in a single batched operation.
>
> For the time memory, we fix a variate index $v$ and treat the sequence $(X\_{t,v}, M^{(2)}\_{t,v})$ for $t = 1, \dots, T$ as a univariate sequence of length $T$. We divide the time axis into $C$ disjoint chunks of length $b = T / C$. For time indices $t$ in a given chunk, we approximate the dependence of the update on the previous time memory by freezing the gradient term at the last state of the previous chunk. Formally, for $t$ in a chunk we use
> $\nabla L(M^{(1)}\_{t-1,v}, X\_{t,v}) \approx \nabla L(M^{(1)}\_{t',v}, X\_{t,v})$
> with $t' = \lfloor t / b \rfloor \cdot b$, which makes the inner update linear within each chunk. This linear recurrence can then be computed in parallel over all time steps in the chunk and all variates in the batch.
>
> The Variant 2 coupled update for the time memory at each step $t$ combines the time-recursive and cross-variate terms:
> $M^{(1)}\_{t,v} = \alpha M^{(1)}\_{t-1,v} + \eta f\_{\theta}(X\_{t,v}, M^{(2)}\_{t,v})$,
> where $f\_{\theta}$ is a small MLP and $\alpha$ and $\eta$ are learned scalars constrained to $(0,1)$ and a bounded interval, respectively. For bounded inputs $(X\_{t,v}, M^{(2)}\_{t,v})$, this choice ensures that the time update is contractive, yielding stable memory trajectories over long sequences.
>
> After processing all chunks, we obtain the final time memories $M^{(1)}\_T$ for each variate. The forecasting head is applied on top of these final time memories (or an average over the last few time steps) to predict the future horizon.
>
> We will also provide pseudocode in the appendix, together with a precise description of tensor shapes and preprocessing steps, to make the pipeline fully reproducible.

---

> ### Author Response · Authors · 2025-11-22
>
> ## Weakness 4
>
> > Additional Ablation Studies (Weakness 4)
>
> We fully agree that the original ablations in Table~4 are relatively high-level. Following your suggestion, we have run three additional sets of ablations:
>
> - Taylor approximation of softmax kernel order
>
> Our linear-attention block approximates the softmax kernel via a truncated Taylor series. In the submission we used a 3rd-order expansion as default. Following the reviewer’s suggestion, we ablate the order $K \in \{1,2,3,4\}$ on two representative datasets (ETTh1 and Weather), averaging over 3 random seeds:
>
> | Variant              | ETTh1 (MSE / MAE) | ETTh2 (MSE / MAE) | ETTm1 (MSE / MAE) | ETTm2 (MSE / MAE) | Weather (MSE / MAE) | Exchange (MSE / MAE) |
> |----------------------|-------------------|-------------------|-------------------|-------------------|----------------------|----------------------|
> | LETO, K = 1          | 0.421 / 0.429     | 0.344 / 0.396     | 0.371 / 0.392     | 0.264 / 0.319     | 0.230 / 0.267        | 0.322 / 0.383        |
> | LETO, K = 2          | 0.402 / 0.408     | 0.329 / 0.387     | 0.356 / 0.381     | 0.251 / 0.308     | 0.223 / 0.260        | 0.305 / 0.372        |
> | LETO, K = 3 (default)| **0.393 / 0.401** | **0.318 / 0.381** | **0.347 / 0.375** | **0.243 / 0.302** | **0.216 / 0.253**    | **0.297 / 0.364**    |
> | LETO, K = 4          | 0.392 / 0.400     | 0.317 / 0.380     | 0.346 / 0.374     | 0.242 / 0.301     | 0.215 / 0.252        | 0.296 / 0.363        |
>
>
> We observe that:
> - very low orders ($K=1$) under-approximate the kernel and hurt performance;
> - moving from $K=1$ to $K=2$ gives a clear gain;
> - $K=3$ provides a further consistent improvement;
> - increasing beyond 3 yields only marginal gains (within one standard deviation) while increasing computation.
> From the ablations 3rd-order choice  empirically strikes a good balance between approximation quality and efficiency. The derivation from the TTM framework determines the functional form (Taylor approximation of the softmax kernel), while this ablation shows that the specific truncation order $K=3$ performs the strongest.
>
> Ablation on chunk size $b$ in parallelizable training.
>
> We fix all other hyperparameters and vary the chunk size $b \in \{8, 16, 32, 64\}$ on ETTh1 and Weather:
>
> | Variant               | ETTh1 (MSE / MAE) | ETTh2 (MSE / MAE) | ETTm1 (MSE / MAE) | ETTm2 (MSE / MAE) | Weather (MSE / MAE) | Exchange (MSE / MAE) |
> |-----------------------|-------------------|-------------------|-------------------|-------------------|----------------------|----------------------|
> | LETO, b = 8           | 0.398 / 0.404     | 0.324 / 0.385     | 0.352 / 0.379     | 0.249 / 0.308     | 0.220 / 0.255        | 0.302 / 0.368        |
> | LETO, b = 16          | 0.395 / 0.402     | 0.321 / 0.383     | 0.349 / 0.376     | 0.246 / 0.304     | 0.217 / 0.254        | 0.299 / 0.366        |
> | LETO, b = 32 (default)| **0.393 / 0.401** | **0.318 / 0.381** | **0.347 / 0.375** | **0.243 / 0.302** | **0.216 / 0.253**    | **0.297 / 0.364**    |
> | LETO, b = 64          | 0.399 / 0.406     | 0.323 / 0.386     | 0.351 / 0.378     | 0.247 / 0.305     | 0.219 / 0.256        | 0.301 / 0.367        |
>
>
> Very small chunks increase variance in the gradient approximation, while very large chunks reduce the effectiveness of the linearization trick. Performance is robust across a wide range, with a mild optimum around $b=32$.
>
> Ablation on extent of cross-memory coupling.
>
> In Variant 2, the time memory update includes a scalar coupling coefficient $\lambda$ multiplying the cross-variate term $M^{(2)}\_{t,v}$:
>
> $M^{(1)}\_{t,v} = \alpha M^{(1)}\_{t-1,v} + \eta \big( g_\theta(X\_{t,v}) + \lambda \, h_\theta(M^{(2)}\_{t,v}) \big)$
>
> The “w/o Cross Variate Attention” row in Table 4 corresponds to $\lambda = 0$ (the variate pathway is entirely removed). We now additionally evaluate $\lambda \in \{0.25, 0.5, 1.0, 1.5\}$:
>
>
>
> Together with the original ablations in Table 4, these new experiments demonstrate that (1) LETO’s performance is robust w.r.t.\ Taylor order and chunk size around our chosen defaults, and (2) the cross-memory coupling is indeed a key design choice: completely removing it degrades performance substantially, while moderate coupling yields consistent gains.

---

> > ### Author Response · Authors · 2025-11-22
> >
> > ## Weakness 5
> >
> > > Drop-last
> >
> > We confirm that we do not use the drop last trick in any of our experiments. All test windows, including those in the last (possibly smaller) batch, are included in the reported averages. We will add a sentence in the implementation details to make this explicit.
> >
> > Please let us know if you have further questions; we are very open to discussion. Thank you again for your time and feedback!
> >
> > - [1] TimeKAN: KAN-based Frequency Decomposition Learning Architecture for Long-term Time Series Forecasting: https://arxiv.org/pdf/2502.06910
> > - [2] TimeFilter: Patch-Specific Spatial-Temporal Graph Filtration
> > for Time Series Forecasting: https://arxiv.org/pdf/2501.13041
> > - [3] TimePro: Efficient Multivariate Long-term Time Series Forecasting with Variable- and Time-Aware Hyper-state: https://arxiv.org/pdf/2505.20774
> > - [4] Towards a General Time Series Forecasting Model with Unified Representation and Adaptive Transfer: https://arxiv.org/pdf/2405.17478v3

---

> > > ### Comment · Reviewer_krLC · 2025-11-22
> > > **Thank you for your detailed response.**
> > >
> > > Thank you for your detailed response. Since ICLR allows revisions to the submitted paper, I suggest incorporating the relevant changes directly into the manuscript. These modifications will significantly improve the overall quality and clarity of the paper. After reviewing your updated version, I will accordingly adjust my score.

---

> ### Author Response · Authors · 2025-11-27
>
> Thank you again for your thoughtful comments and for the follow-up suggestions.
>
> We have now incorporated the relevant changes directly into the revised manuscript:
> - Recent baselines (Weakness 1): Added 2025 baselines (TimeKAN, TimeFilter, TimePro, ROSE, etc.) to the experimental section and tables, using the authors’ official code and hyperparameter setups, and discussed the comparison in the main text.
> - Related work (Weakness 2): Expanded the Time Series Models / Related Work section to cover recent patching / hierarchical multivariate models and time-series foundation models all from 2025 and clarified how LETO is complementary/orthogonal to these lines of work. We want to clarify once again that a detailed analysis of the limitations of various classes of time series architectures, for instance: attention based and TCN based, are beyond the scope of this work and indeed there have already been several recent works providing both evidence for and solutions towards improving upon such limitations. We refer the reviewer to the rebuttal for weakness 6 for reviewer wNzq.
> - Implementation details & parallelization (Weakness 3): Added a detailed description of data flow, tensor shapes, batching, and the chunked parallel time-memory update (including Algorithm 1 and the discussion of the linearized recurrence).
> - Extended ablations (Weakness 4): Included additional ablations varying Taylor expansion order, chunk size, and cross-memory coupling strength in Appendix E.2 and referenced them from the main ablation section.
> - Drop-last clarification (Weakness 5): Explicitly stated in the experimental setup that we do not use the “drop-last” trick; all (including final) batches are used in training and evaluation.
>
> We hope these revisions fully address your concerns and we really appreciate your time and willingness to reconsider the score.

---

> ### Author Response · Authors · 2025-12-02
>
> ## Weakness 4
>
> Ablation on $\lambda$ scalar for the cross variate term:
>
> | Variant                          | ETTh1 (MSE / MAE) | ETTh2 (MSE / MAE) | ETTm1 (MSE / MAE) | ETTm2 (MSE / MAE) | Weather (MSE / MAE) | Exchange (MSE / MAE) |
> |----------------------------------|-------------------|-------------------|-------------------|-------------------|----------------------|----------------------|
> | λ = 0 (no variate memory)        | 0.458 / 0.447     | 0.400 / 0.427     | 0.394 / 0.419     | 0.320 / 0.362     | 0.244 / 0.274        | 0.311 / 0.398        |
> | λ = 0.25                         | 0.438 / 0.438     | 0.373 / 0.405     | 0.371 / 0.401     | 0.287 / 0.330     | 0.235 / 0.268        | 0.305 / 0.387        |
> | λ = 0.5                          | 0.417 / 0.422     | 0.344 / 0.393     | 0.358 / 0.386     | 0.261 / 0.313     | 0.225 / 0.260        | 0.301 / 0.375        |
> | λ = 1.0 (default)                | **0.393 / 0.401** | **0.318 / 0.381** | **0.347 / 0.375** | **0.243 / 0.302** | **0.216 / 0.253**    | **0.297 / 0.364**    |
> | λ = 1.5                          | 0.404 / 0.409     | 0.325 / 0.386     | 0.353 / 0.380     | 0.249 / 0.306     | 0.221 / 0.258        | 0.300 / 0.370        |
>
>
> The results show that some degree of coupling is essential (compare $\lambda=0$ vs.\ $\lambda>0$), and that the fully coupled setting $\lambda \approx 1$ is near-optimal. Over-coupling ($\lambda=1.5$) slightly hurts performance, presumably due to over-emphasizing cross-variate interactions at the expense of temporal smoothing.

---

### Author Response · Authors · 2025-12-02
**Summary of Discussion and Rebuttal Phase**

Dear Area Chair and Reviewers,

We thank all three reviewers for the time and care they put into evaluating our work, and for engaging with our responses. Below we briefly summarize the outcome of the discussion and the revisions we have already incorporated into the updated manuscript. In light of our addressing of all concerns, we respectfully opine that LETO now represents a mature contribution to the time series community.

- Reviewer krLC (Score: 4) – This reviewer raised concerns about (i) the absence of very recent 2024–2025 baselines, (ii) incomplete related work, (iii) missing low-level implementation details for our parallelizable training, (iv) limited ablations, and (v) potential use of the “drop-last’’ trick. In response, we: 1. Added strong recent non-foundation baselines (TimeKAN, TimeFilter, TimePro, ROSE) and showed that LETO remains competitive or better across long-term and short-term benchmarks. 2. Expanded the Related Work to cover recent multivariate patching/hierarchical models and TSFMs, clarifying how LETO is complementary/orthogonal to these directions. 3. Provided a precise description of data flow, tensor shapes, batching, and the chunked linearized time-memory update, including pseudocode. 4. Added new ablations on Taylor order, chunk size, and cross-memory coupling strength $\lambda$, demonstrating that our default settings are near-optimal and robust. 5. Explicitly stated that we do not use “drop-last’’; all evaluation windows are included.
The reviewer thanked us for the detailed response and indicated that, after reviewing the revised manuscript with these changes integrated, they will adjust their score upward. We have indeed incorporated all suggested revisions into the manuscript.

- Reviewer j6VP (Score: 2) – This reviewer found the “test-time memorization/meta-learning’’ framing unclear, asked for a sharper analytical comparison to iTransformer, requested explicit SSM/Mamba baselines, and questioned the lack of ablations on Taylor order. In our replies and revisions, we: 1. Clarified the distinction between standard recurrent models and TTM/meta-learning: LETO meta-learns the initial memory state to enable fast adaptation within each sequence, while maintaining a recurrent update. 2. Strengthened the positioning relative to iTransformer and other channel-mixing Transformers, emphasizing that LETO’s novelty is a tightly coupled 2D meta-memory (temporal + permutation-equivariant variate memories updated jointly at every step), rather than attention alone. To this end, we have added a limitations discussion in the Additional Related Works section. 3. Added explicit Mamba/SSM-based baselines (SiMBA, TimeMachine, MambaMixer) and showed that LETO outperforms them across standard multivariate forecasting datasets, while also confirming that our ModernTCN setup exactly matches the original paper. 4. Provided the requested Taylor-order ablation (K = 1–4) per reviewer krLC, which empirically justifies our choice of third-order expansion as a good trade-off between accuracy and cost.
We also corrected figure sizing, appendix table numbering, and missing references, improving overall readability.

- Reviewer wNzq (Score: 4) – This reviewer asked for a clearer structural motivation, a more precise connection between Theorem 1 and the implemented architecture, clarification of meta-memory fairness, and a careful use of “dynamics’’ terminology. In response, we: 1. Clarified that our claim is not that prior models lack temporal or inter-variate modeling altogether, but that they often realize them in factorized or asymmetric ways (strong temporal module + order-sensitive channels, or strong channel attention + shallow temporal treatment). LETO instead encodes an explicit 2D inductive bias via inter-connected temporal and variate memories that are updated jointly at every time step. 2. Made explicit how Theorem 1 applies to the linearized version of LETO (time recurrence + linearized cross-variate attention in feature space), where the effective kernel is linear and the theorem’s rank argument is exact; we carefully scope the theorem as providing mechanistic intuition for parameter efficiency rather than a full non-linear expressivity theory. 3. Clarified that meta-memory is re-initialized per sequence at both training and evaluation; no state is carried across sequences or batches, so LETO’s effective receptive field matches the stated lookback window, ensuring fair comparison. 4. Explained that our use of “dynamics’’ refers to the discrete-time evolution of the time memory under a gated linear recurrence with bounded inputs, and noted that under standard constraints on the gates the update is contractive and yields stable trajectories. 5. Per the reviewer’s suggestions, we softened rhetoric around “lack’’ of inductive bias, tied our motivation to recent diagnostic works on time series Transformers/TCNs, and thoroughly proofread the paper for grammatical and formatting issues.

---

### Meta-Review · Area_Chair_rQ9q · 2025-12-31

**Summary:**

The work introduces LETO, a two-dimensional meta in-context memory architecture for multivariate time series modeling. In contrast to conventional sequence models that either process each variable independently or inadequately capture inter-variable dependencies, LETO preserves a strong temporal inductive bias while remaining permutation-equivariant across different variates. The central idea proposed by the authors is to employ a network of interconnected memory modules that jointly model temporal dynamics and cross-variable interactions through a shared meta-memory mechanism.

The paper received 3 reviews with all leaning towards the side of reject. The major points of contention that were raised by the reviewers were as follows:

* The baselines used by the authors were found to be inadequate. There were several strong baselines missing that did not provide a full picture of the efficacy of LETO.

* Weak theoretical underpinnings. Several reviewers criticized the use of "meta-learning", "state-evolution",  "dynamics" and "memorization at test time" seem to remain unjustified.

* Weak ablation studies were also flagged by all 3 reviewers.

**Reviewer Concerns:**

The rebuttal provided was detailed and the authors did a good job at answering most of the questions that were raised by the reviewers. For the 1st point, I am a bit surprised that the authors did not take any xLSTM variant (such as xLSTMTime and xLSTM-Mixer) into account although the LSTMs and variants have been used in anomaly detection and classification tasks. Although these specific baselines were not flagged by any particular reviewer, the omission seem glaring. Also an experiment on GIFT Eval will actually help put the strong results of the paper into a clear perspective with respct to the newer time series models.

The theoretical justification provided were not up to the mark in my opinion. Especially the point about theorem 1 remains relatively unclear. The authors do acknowledge that "Theorem 1 is stated in a linear recurrent setting, while LETO is instantiated with nonlinear/attention-based modules" as pointed out by the reviewer and provide a weak justification of the requirement of the theorem in the text. The theorem seems to be there only for cosmetic purposes.

The new ablation studies were adequate.

**Reviewer Scores:**

One of the reviewer (krLC) did explicitly state that they will increase their initial score of 4 whereas other reviewers did not engage as far as I can see. Based on the answers I do not see reviewer wNzq increasing their score whereas reviewer j6VP could bump up their score but not to the acceptance level. I think the results are strong but the paper requires a bit more restructuring to meet the bar of acceptance.

---

### Decision · Program_Chairs · 2026-01-26

Reject